# HIGH FIDELITY VISUALIZATION OF WHAT YOUR SELF-SUPERVISED REPRESENTATION KNOWS ABOUT

## ABSTRACT

Discovering what is learned by neural networks remains a challenge. In self-supervised learning, classification is the most common task used to evaluate how good a representation is. However, relying only on such downstream task can limit our understanding of how much information is retained in the representation of a given input. In this work, we showcase the use of a conditional diffusion based generative model (RCDM) to visualize representations learned with self-supervised models. We further demonstrate how this model's generation quality is on par with state-of-the-art generative models while being faithful to the representation used as conditioning. By using this new tool to analyze self-supervised models, we can *show visually* that i) SSL (backbone) representation are not really invariant to many data augmentation they were trained on. ii) SSL projector embedding appear too invariant for tasks like classifications. iii) SSL representations are more robust to small adversarial perturbation of their inputs iv) there is an inherent structure learned with SSL model that can be used for image manipulation.

| Earth from . . . space[1] | an untrained representation | a supervised representation | a SSL representation |
| --- | --- | --- | --- |

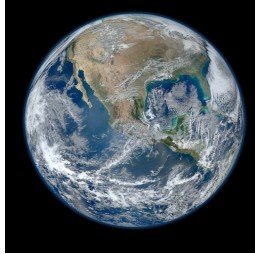 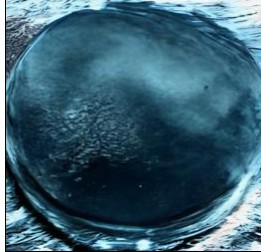 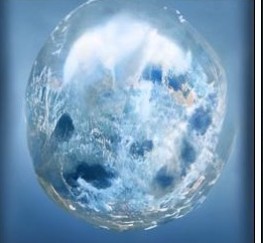 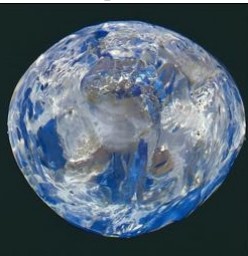

## 1 INTRODUCTION AND MOTIVATION

Approaches aimed at learning useful representations, from unlabeled data, have a long tradition in machine learning. These include probabilistic latent variable models and variants of auto-encoders (Ackley et al., 1985; Hinton et al., 2006; Salakhutdinov et al., 2007; Vincent et al., 2008; Kingma & Welling, 2014; Rezende et al., 2014), that are traditionally put under the broad umbrella term of *unsupervised learning* (Bengio et al., 2013). More recent approaches, under the term of *self-supervised learning* (SSL) have used various kinds of "pretext-tasks" to guide the learning of a useful representations. Filling-in-the-blanks tasks, proposed earlier in (Vincent et al., 2008; 2010), later proved remarkably successful in learning potent representations for natural language processing (Vaswani et al., 2017; Devlin et al., 2019). Pretext tasks for the image domain include solving Jigsaw-puzzles (Noroozi & Favaro, 2016), predicting rotations or affine transformations (Gidaris et al., 2018; Zhang et al., 2019b) or discriminating instances (Wu et al., 2018; van den Oord et al., 2018). The latest, most successful, modern family of SSL approaches for images (Misra & Maaten, 2020; Chen et al., 2020; Chen & He, 2020; He et al., 2020; Grill et al., 2020; Caron et al., 2020; 2021; Zbontar et al., 2021; Bardes et al., 2021), have two noteworthy characteristics that markedly distinguish them from

---

[1]We use representations of the real picture of Earth on the left (source: NASA) as conditioning for RCDM. We show samples (resolution $256 \times 256$) in cases where the representations (2048-dimensions) were obtained respectively with a random initialized ResNet50, a supervised-trained one, and a SSL-trained one. More samples in Fig. 34.

traditional unsupervised-learning models such as autoencoder variants or GANs (Goodfellow et al., 2014): a) their training criteria are not based on any input-space reconstruction or generation, but instead depend only on the obtained distribution in the representation or embedding space b) they encourage invariance to explicitly provided input transformations a.k.a. data-augmentations, thus injecting important additional domain knowledge.

Despite their remarkable success in learning representations that perform well on downstream classification tasks, rivaling with supervised-trained models (Chen et al., 2020), much remains to be understood about SSL algorithms and the representations they learn. How do the particularities of different algorithms affect the representation learned and its usefulness? What information does the learned representation contain? Empirical analyses have so far attempted to analyse SSL algorithms almost exclusively through the limited lens of the numerical performance they achieve on downstream tasks such as classification. Contrary to their older unsupervised learning cousins, due to characteristic a) highlighted above, modern SSL methods do not provide any direct way of mapping back the representation in image space, to allow *visualizing* it. The main goal of our work is thus to enable the visualization of representations learned by SSL methods, as a tool to improve our understanding.

More precisely, we suppose that we are given a mapping function $f$ – a (part of) a SSL or otherwise trained neural network – that takes an input image $\boldsymbol{x} \in \mathbb{X}$ and maps it to a representation $\boldsymbol{h} \in \mathbb{H}$ as in $\boldsymbol{h} = f(\boldsymbol{x})$. The input space $\mathbb{X}$ will typically be RGB pixel space represented as $\mathbb{X} = [-1, 1]^D$, and the representation space $\mathbb{H}$ will be the output space of a deeper layer. We denote the representation space's dimension by $K$ as in $\mathbb{H} = \mathbb{R}^K$. Now we want, when given a specific representation $\boldsymbol{h} \in \mathbb{H}$, to visualize what inputs $\boldsymbol{x}$ yield this representation. As $f$ is typically *not* bijective, e.g. if it computes a higher level representation of reduced dimension, there may be many inputs that yield that same representation, most of which will not resemble natural images. Our approach (Section 3) thus aims at finding inputs that not only map to the target $\boldsymbol{h}$ but are also visually recognizable images. For this we build a conditional generative model that (implicitly) models $p(\boldsymbol{x}|\boldsymbol{h})$ and allows to sample diverse $\boldsymbol{x}' \sim p(\boldsymbol{x}|\boldsymbol{h})$. For reasons that we will explain later, we opted for a conditional diffusion model, inspired by Dhariwal & Nichol (2021), for our conditional generative model.

This paper's main contributions are:

- To devise a conditional diffusion model architecture (RCDM) suitable for conditioning on large vector representations s.a. SSL representations. Our model provides high-quality images, measured in term of FID, on par with state-of-the-art models (Tab. 2a), and is suited for out-of-distribution samples (see Fig. 1). The conditionally generated images are also highly representation-faithful i.e. they closely match the representations of the images used for the conditioning (Tab. 2b, Fig. 24).
- To showcase its usefulness for qualitatively analyzing SSL representations and embeddings (also in contrast with supervised representations), by shedding light on what information about the input image is or isn't retained in them.

Specifically, by repeatedly sampling from a same conditioning representation, one can observe which aspects are common to all samples, thus identifying what is encoded in the representation, while the aspects that vary greatly show what was *not retained* in the representation. We make the following observations: (i) SSL projector embeddings appear most invariant, followed by supervised-trained representation and last SSL representations[2] (Fig. 3). (ii) SSL-trained representations retain more detailed information on the content of the background and object style while supervised-trained representations appear oblivious to these (Fig. 4). (iii) despite their invariant training criteria, SSL representations appear to retain information on object scale, grayscale vs color, and color palette of the background, much like supervised representation (Fig. 4). (iv) Supervised representations appear more susceptible to adversarial attacks than SSL ones (Fig. 5,30). (v) We can explore and exploit structure inside SSL representations leading to meaningful manipulation of image content (s.a. splitting representation in foreground/background components to allow background substitution) (Fig, 6, 31, 32).

## 2 RELATED WORK

**Deterministic visualization methods:** Many early works (Erhan et al., 2009; Zeiler & Fergus, 2014; Simonyan et al., 2013; Selvaraju et al., 2016; Smilkov et al., 2017) used gradient based

---

[2]The representation that is produced by a Resnet50 backbone, before the projector.

techniques to visualize what is learned by neural networks. This led to successful interpretability of DNs internal features, especially when applied on a unit belonging to the first few layers of a DN (Cadena et al., 2018). More recently, Caron et al. (2021) used the attention mask of transformers to perform unsupervised object segmentation. By contrast, our method is not model dependent, we can plug any type of representation as conditioning for the diffusion model. Another possibility, explored in Zhao et al. (2021); Appalaraju et al. (2020); Ericsson et al. (2021), is to learn to invert the DN features through a Deep Image Prior (DIP) $g_\theta$ as in $\min_\theta d(g_\theta(f(\boldsymbol{x})), \boldsymbol{x})$. In fact, as we experimented in Appendix A, performing unconstrained gradient based optimization of a sample to match a target representation leads to unrealistic generation. The use of DIP however requires to retrain the DIP network for each feature-image pairs and only quantify how much information about $\boldsymbol{x}$ is retained in $f(\boldsymbol{x})$ while we are interested in finding all the $\boldsymbol{x}$s that are seen to have the same information content.

**Generative models:** Several families of techniques have been developed as generative models, that can be trained on unlabeled data and then employed to generate images. These include auto-regressive models (Van Den Oord et al., 2016), variational auto-encoders (Kingma & Welling, 2014; Rezende et al., 2014), GANs (Goodfellow et al., 2014), autoregressive flow models (Kingma et al., 2016), and diffusion models (Sohl-Dickstein et al., 2015). Conditional versions are typically developed shortly after their unconditional versions (Mirza & Osindero, 2014; van den Oord et al., 2016). In principle one could envision training a conditional model with any of these techniques, to condition on an SSL or other representation for visualization purpose, as we are doing in this paper with a diffusion model. One fundamental challenge when conditioning on a rich representation such as the one produced by a SSL model, is that for a given conditioning $\boldsymbol{h}$ we will usually have available only a *single* corresponding input instance $\boldsymbol{x}$, precious few to learn a distribution. This can lead model training astray. By contrast a particularly successful model such as the conditional version of BigGAN (Brock et al., 2019) conditions on a categorical variable, the class label, that for each value of the conditioning has a large number of associated $\boldsymbol{x}$ data.

One closely related work to ours is the recent work on Instance-Conditioned GANs (IC-GAN) of Casanova et al. (2021). Similar to us it also uses SSL or supervised representations as conditioning when training a conditional generative model, here a GAN (Goodfellow et al., 2014), specifically a variant of BigGAN (Brock et al., 2019) or StyleGAN2 (Karras et al., 2020). However, the model is trained such that, from a specific representation $\boldsymbol{h}$, it learns to generate not only images that should map to this representation, but a much broader neighborhood of the training data. Specifically up to 50 training points that are nearest neighbors in representation space to $\boldsymbol{h}$. It remains to be seen whether such a GAN architecture could be trained successfully without resorting to a nearest neighbor set. IC-GAN is to be understood as a conditional generative model of an image's broad *neighborhood*, and the primary focus of this work was on developing a superior quality controllable generative model. By contrast we want to sample images that map as closely as possible to the original image in the representation space, as our focus is to build a tool to analyse SSL representations, to enable visualising what images correspond *precisely* to a representation. (See Fig. 24 for a comparison.)

As previously stated, our choice of a diffusion-based model rather than a GAN was motivated by the simple stable training of such models, by the high quality of generated images demonstrated with the model we build on Dhariwal & Nichol (2021) that rivals that of GAN, and by the similarity of the input-space gradient-based sampling procedure with the simple approach we explored in Appendix A. While conditional versions of their diffusion model were already developed in Dhariwal & Nichol (2021), these were unsuitable for conditioning on high dimensional distributed representation s.a. those obtained with SSL models, as we discussed in details in section 3. This prompted us to develop the architecture variant of this paper. Despite their qualities, diffusion models also have drawbacks, in particular they are resource-hungry and slow for generation. It is thus very likely that alternative approaches for representation-conditioned generative models will be developed and employed for analysis and visualisation purposes in the future.

Lastly, a few approaches have focused on conditional generation to unravel the information encoded in representations of supervised models. In Shocher et al. (2020), a hierarchical LSGAN generator is trained with a class-conditional discriminator (Zhang et al., 2019a). While the main applications focused on inpainting and style-transfer, this allowed to visually quantify the increasing invariance of representations associated to deeper and deeper layers. This method however requires labels to train the generator. On the other hand, Nash et al. (2019) proposed to use an autoregressive model, in particular PixelCNN++ (Salimans et al., 2017), to specifically study the invariances that each layer

of a DN inherits. In that case, the conditioning was incorporated by regressing a context vector to the generator biases. As far as we are aware, PixelCNN++ generator falls short on high-resolution images e.g. most papers focus on $32 \times 32$ Imagenet. Lastly, Rombach et al. (2020) proposes to learn a Variational AutoEncoder that is combined with an invertible neural network (INN) whose role is to model the relation between the VAE latent space and the given representations. To allow for interpretable manipulation, a second invertible network (Esser et al., 2020) is trained using labels to disentangle the factors of variations present in the representation. By contrast we train end-to-end a single decoder to model the entire diversity of inputs that correspond to the conditioning representation, without imposing constraints of a structured prior or requiring labels for image manipulation.

## 3 CONDITIONING A DIFFUSION MODEL ON REPRESENTATION $h$

We propose to build a novel conditional diffusion process whose goal is to directly generate realistic images that match a given target representation. Given a representation $h$ the failure of the method in Appendix A suggests we need a way to further constrain the type of samples we generate, beyond the mere constraint of belonging to $\mathcal{S}(h)$. More precisely, we want to be able to sample, among $\mathcal{S}(h)$, inputs that are more like the training data (here natural images), i.e. that are likely under the same distribution. That is we would like not merely to find $x' \in \mathcal{S}(h)$ but rather to sample $x' \sim p(x|h)$. Informally we might picture the *set* of likely natural images (points whose density $p(x)$ is above some threshold) as a subset of $\mathbb{X}$ that we will loosely refer to as the "data manifold" $\mathcal{M}$. Where our first approach attempted to sample points more or less uniformly within $\mathcal{S}(h)$, modeling and sampling form $p(x|h)$ will more likely produce points from $\mathcal{M} \cap \mathcal{S}(h)$. We propose to train a *conditional diffusion model* to implicitly model $p(x|h)$ and allow sampling from it.

While we could have considered other conditional generative approaches (we discussed some of the alternatives in section 2), the choice of the reverse diffusion approach (Sohl-Dickstein et al., 2015; Ho et al., 2020; Song & Ermon, 2019; 2020; Song et al., 2021a;b; Nichol & Dhariwal, 2021) is not entirely arbitrary. It is motivated by the remarkable quality of image generation they recently proved capable of (Dhariwal & Nichol, 2021), as well as the closeness of their Langevin-MCMC-like generation process to the input-gradient-directed optimization we used to obtain samples in Appendix A. Indeed sampling from an reverse diffusion model similarly starts from a random noise image, and takes multiple steps in input space that can be thought of as (noisy) gradient steps on an (implicit) energy function (Ho et al., 2020; Song et al., 2021a;b) Informally, one can think of these steps as progressively moving this initial random point closer to the "data manifold" $\mathcal{M}$. A reverse diffusion *conditioned* on $h$ will move it towards $\mathcal{M} \cap \mathcal{S}(h)$. The three sampling approaches are depicted and contrasted in Fig. 11a.

We base our work on the Ablated Diffusion Model (ADM) developed by Dhariwal & Nichol (2021) which uses a UNet architecture (Ronneberger et al., 2015) to learn the reverse diffusion process.Our conditional variant – called *Representation-Conditionned Diffusion Model* (RCDM) – is illustrated in Fig. 11b. To suitably condition on representation $h = f(x)$, we replaced the Group Normalization layers of ADM by conditional batch normalization layers (Dumoulin et al., 2017) that take $h$ as conditioning[3]. More precisely we apply a fully connected layer to $h$ that reduces dimension to a vector of size 512. This vector is then given as input to multiple conditional batch normalization layers that are placed in each residual block of the diffusion model.

In contrast with Dhariwal & Nichol (2021) we don't use the input gradient of a classifier to bias the reversed diffusion process towards more probable images, nor do we use any label information for training our model – recall that our goal is building a visualization tool for SSL models that train on unlabeled data. Our batch normalization based conditioning is also different from the approach that was used by Dhariwal & Nichol (2021) when conditioning their super-resolution model on a low-resolution image. Their technique of upscaling and appending the conditioning image as extra channels to the input would not work for our application. Our representation $h$ typically has 2048 "channels" with no spatial extent: upscaling it to the size of the input image would blow up memory constraints.

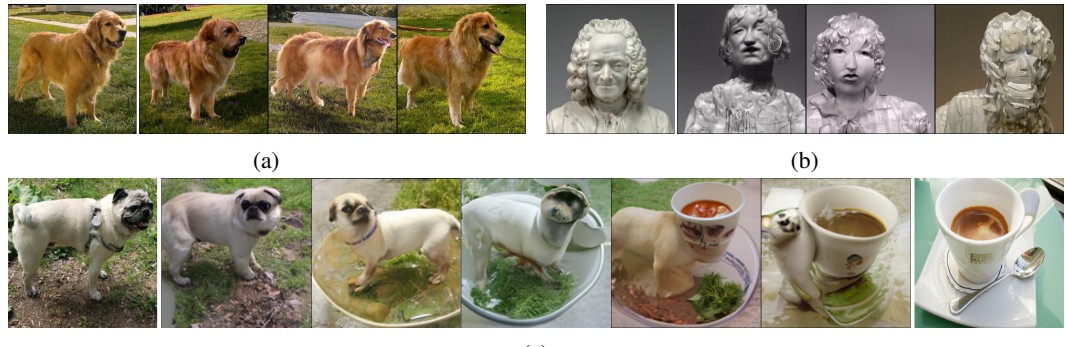

(a)                         (b)

(c)

Figure 1: a) In distribution conditional image generation. An image from ImageNet validation set (first column) is used to compute the representation output by a trained SSL model (Dino). The representation is used as conditioning for the diffusion model. Resulting samples are shown in the subsequent columns (see Fig. 12). We observe that our conditional diffusion model produces samples that are very close to the original image. b) Out of distribution (OOD) conditional. How well does RCDM generalize when conditioned on representations given by images from a different distribution? (here a WikiMedia Commons image, see Fig. 13 for more). Even with OOD an conditionning, the images produced by RCDM are very close visually to the original image. c) Interpolation between two images from ImageNet validation data. We apply a linear interpolation between the SSL representation of the images on the first column and the representation of the images on the last column. We use the interpolated vector as conditioning for our model that produce the samples that are showed in column 2 to column 6. Fig. 16 in appendix shows more sampled interpolation paths.

(a) We report results for ImageNet to show that our approach is reliable for generating images which look realistic. Since the focus of our work is not generative modelling but to showcase and encourage the use of such model for representation analysis, we only show results for one conditional generative models. For each method, we computed FID and IS with the same evaluation setup in Pytorch.

(b) For each encoder, we compute the rank and mean reciprocal rank (MRR) of the image used as conditioning within the closest set of neighbor in the representation space of the samples generated from the valid set (50K samples). A rank of one means that all of the generated samples for a given model have their representations matching the representation used as conditioning.

| Method | Res. | ↓FID | ↑IS |
|---|---|---|---|
| ADM (Dhariwal & Nichol, 2021) | 256 | 26.8 | 34.5 ± 1.8 |
| IC-GAN (Casanova et al., 2021)) | 256 | 20.8 | 51.3 ± 2.2 |
| IC-GAN (Casanova et al., 2021) (KDE*) | 256 | 21.6 | 38.6 ± 1.1 |
| **RCDM (ours)** | 256 | 19.0 | 51.9 ± 2.6 |

| Model | ↓Mean rank | ↑MRR |
|---|---|---|
| Dino (Caron et al., 2021) | 1.00 | 0.99 |
| Swav (Caron et al., 2020) | 1.01 | 0.99 |
| SimCLR (Chen et al., 2020) | 1.16 | 0.97 |
| Barlow T. (Zbontar et al., 2021)) | 1.00 | 0.99 |
| Supervised | 5.65 | 0.69 |

Figure 2: a) Table of results on ImageNet. We compute the FID (Heusel et al., 2017) and IS (Salimans et al., 2016) on 10 000 samples generated by each models with 10 000 images from the validation set of ImageNet as reference. KDE* means that we used our unconditional representation sampling scheme based on KDE (Kernel Density Estimation) for conditioning IC-GAN instead of the method based on K-means introduces by Casanova et al. (2021). b) Table of ranks and mean reciprocal ranks for different encoders. This table show that RCDM is faithful to the conditioning by generating images which have their representations close to the original one.

## 4    EXPERIMENTS USING RCDM TO MAP BACK REPRESENTATIONS TO IMAGES

Our first experiments aim at evaluating the abilities of our model to generate realistic-looking images whose representations are close to the conditioning. To do so, we trained our Representation-Conditionned Diffusion Model (RCDM), conditioned on the 2048 dimensional representation given by a Resnet50 (He et al., 2016) trained with Dino (Caron et al., 2021) on ImageNet (Russakovsky et al., 2015). Then we compute the representations of a set of images from ImageNet validation data to condition the sampling from the trained RCDM. Fig. 1a shows it is able to sample images that are very close visually from the one that is used to get the conditioning. We also evaluated the generation abilities of our model on out of distribution data. Fig. 1b shows that our model is able to sample new

---

[3]A similar technique was used by Casanova et al. (2021) for IC-GAN, discussed in the next section.

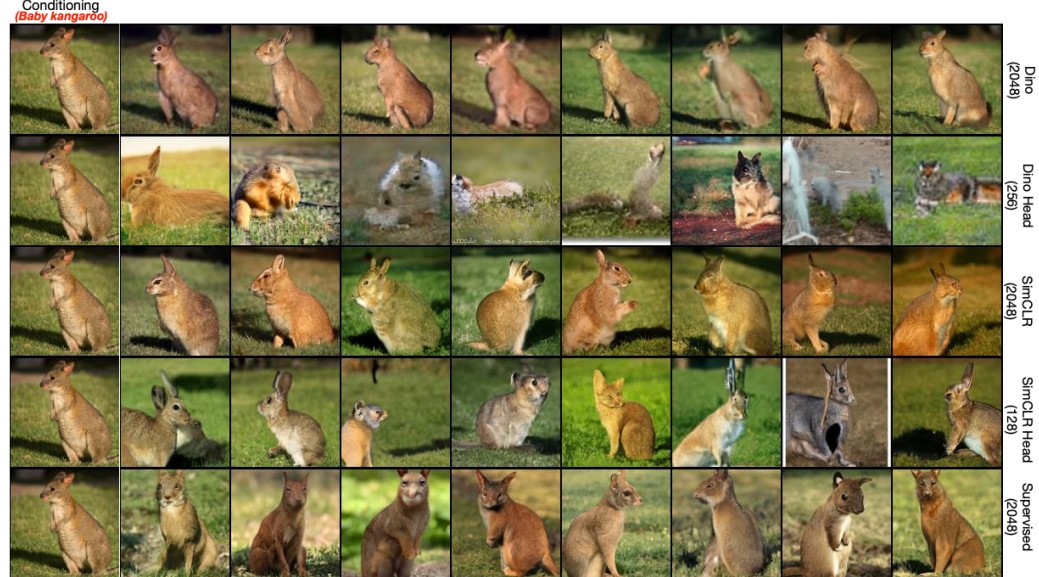

Figure 3: On the first and second row, RCDM samples conditioned on the usual backbone representation (size 2048) and projector representation (size 256) with Dino. Same on the third and forth row with SimCLR representations (2048 and 128). For comparison, we also added samples from RCDM trained on representation given by a supervised trained model. We clearly observe that the projector only keeps global information and not its context, as opposed to the backbone. This indicates that invariances in SSL models are mostly achieved in the projector representation, not the backbone. Additional comparisons provided in Fig. 25,26,12.

views of an OOD image. We also quantitatively validate that the generated images' representations are close to the original image representation in Tab. 2b, Fig. 18, Fig. 19 and Fig. 20.

This implies that there is much information kept inside the SSL representation so that the conditional generative model is able to reconstruct many characteristics of the original image. We also perform interpolations between two SSL representations in Fig. 1c. This shows that our model is able to produce interpretable images even for SSL representations that correspond to an unlikely mix of factors. Both the interpolation and OOD generation clearly show that the RCDM model is not merely outputting training set images that it could have memorized. This is also confirmed by Fig. 17 in the appendix that shows nearest neighbors of generated points.

The conditional diffusion model might also serve as a building block to hierarchically build an unconditional generative model. Any technique suitable for modeling and sampling the distribution of (lower dimensional) representations could be used. As this is not our primary goal in the present study, we experimented only with simple kernel density estimation (see appendix for details). This allow us to quantify the quality of our generative process in an unconditional manner to fairly compare against state-of-the-art generative models such as ADM. We provide some generative model metrics in Tab. 2a along some samples in Fig. 12 to show that our method is competitive with the current literature.

## 5 VISUAL ANALYSIS OF REPRESENTATIONS LEARNED WITH SELF-SUPERVISED MODEL

Having generated samples that are close in the representation space to a conditioning image can gives us an insight on what's hidden in the representations learned with self-supervised models. As demonstrated in the previous section, the samples that are generated with RCDM are really close visually to the image used as conditioning. This give an important proof of how much is kept inside a SSL representation. However, it's also important to consider how much this amount of "hidden" information varied depending on the SSL representation that is used. Therefore, we train several RCDM on SSL representations given by VicReg (Bardes et al., 2021), Dino (Caron et al., 2021), Barlow Twins (Zbontar et al., 2021) and SimCLR (Chen et al., 2020). In many applications that used self-supervised models, the representation that is used is the one that corresponds to the backbone of the ResNet50. Usually, the representation given by the projector of the SSL-model (on which the SSL

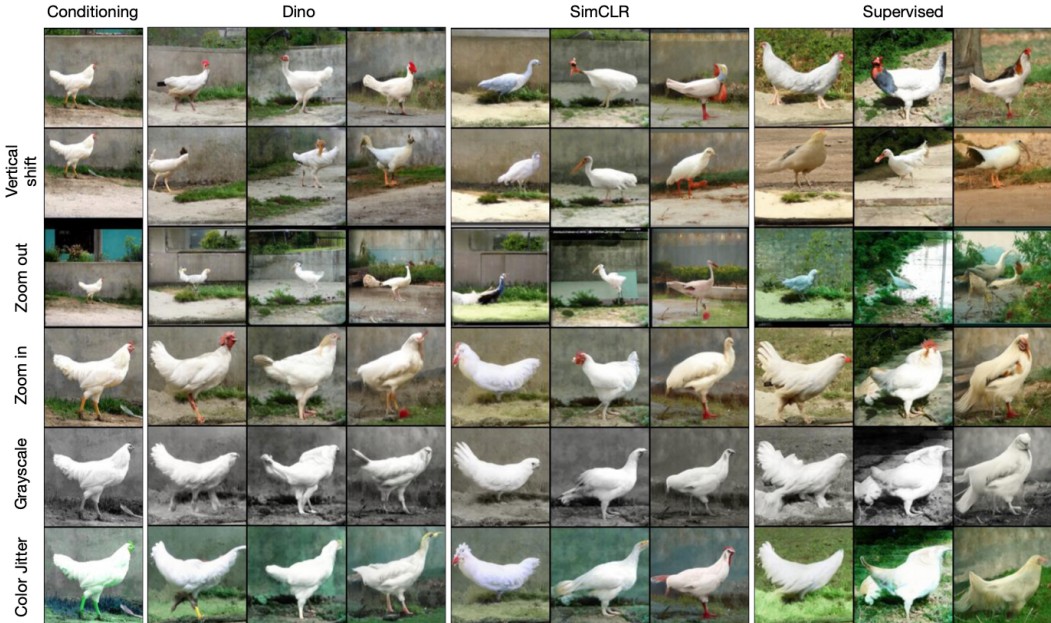

Figure 4: We use our conditional generative model to get insight on the invariance (or covariance) of representations with respect to several data augmentations. On an original image (top left) we apply specific transformations (visible in the first column). For each transformed image, we compute the SSL representation with Dino, SimCLR and a supervised network and condition the corresponding RCDM with that representation to sample 3 images. We see that despite their invariant training criteria, the 2048 dimensional SSL representations appear to retain information on object scale, grayscale vs color, and color palette of the background, much like the supervised representation. They do appear insensitive to vertical shifts. We also see that supervised representation constrain the appearance much less. Refer to Fig. 27 in Appendix for a comparison with using the lower dimensional projector head embedding as the representation.

criterion is applied) is discarded because the results on many downstream tasks like classification is not as good as the backbone. However, since our work is to visualize and better understand the differences between SSL representations, we also trained RCDM on the representation given by the projector of Dino, Barlow Twins and SimCLR. In Fig. 3 and Fig. 25 we condition all the RCDM with the image labelled as conditioning and sample 9 images for each model. We observe that Dino representation does not allow much variance meaning that even information about the pose of the animal is kept inside the representation. In contrast, the SimCLR representation seems to be more invariant to the pose of the kangaroo. We also observe class-crossing, the kangaroo becomes a rabbit. VicReg seems to be more robust in the sense that the animal doesn't cross the class boundary despite changes in the background.

## 5.1 WHAT ARE REPRESENTATIONS REALLY INVARIANT TO?

In Fig. 4, we apply specific transformations (augmentations) to a test image and we check whether the samples generated by the diffusion model change accordingly. We also compare with the behavior of a supervised model. We note that despite their invariant training criteria, the 2048 dimensional SSL representations do retain information on object scale, grayscale status, and color palette of the background, much like the supervised representation. They do appear invariant to vertical shifts. In the Appendix, Fig. 27 applies the same transformations, but additionally compares using the 2048 representation with using the lower dimensional projector head embedding as the representation. There, we observe that the projector representation seems to encode object scale, but contrary to the 2048 representation, it appears to have gotten rid of grayscale-status and background color information. Currently, researchers need to use custom datasets (in which the factors of variation of a specific image are annotated) to verify how well the representations learned are invariant to those

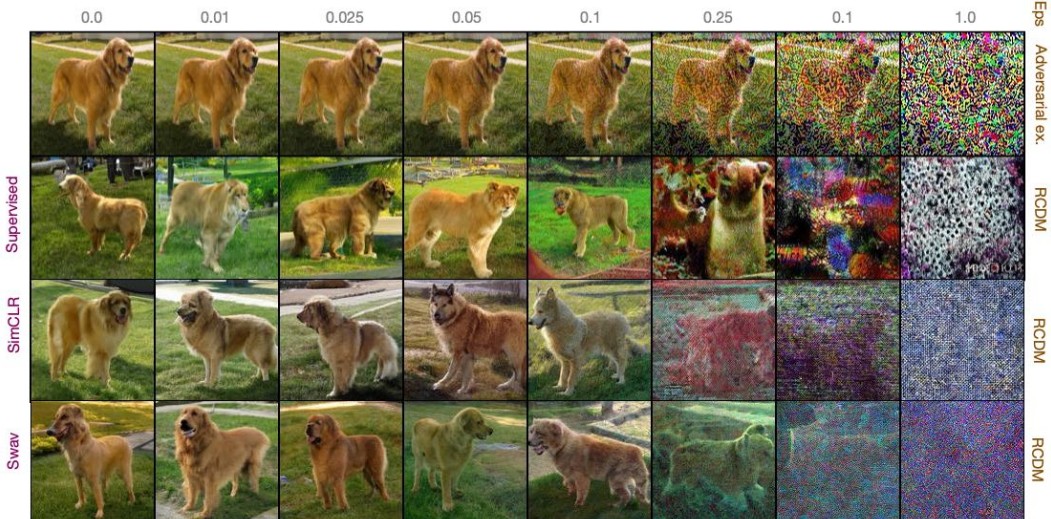

Figure 5: Visualization of adversarial attacks with RCDM. We use Fast Gradient Sign to attack a given image (top-left corner) with various values for the attack coefficient epsilon. On the first row, we only show the adversarial images obtained from a supervised encoder: refer to Fig. 30 in the Appendix to see the (similar looking) adversarial examples for each model. On the following rows, we show the reconstruction of the attacked images with RCDM for various models. For a supervised representation, RCDM reconstructs an animal that belong to another class, a lion in this case. However, we observe that if we use SimCLR or Swav as encoder (third and forth row), the images generated by RCDM are still dogs even with higher values for epsilon.

factors. We hope that RCDM will help researchers in self-supervised learning to alleviate this concern since our method is "plug and play" and can be use on any dataset with any type of representation.

## 5.2 Visualization of adversarial examples

Since our model is able to project any representation to the manifold of real images, we follow the same experiment protocol as Rombach et al. (2020) to visualize how adversarial examples are seen by RCDM. We apply Fast Gradient Sign attacks (FGSM) (Goodfellow et al., 2015) over a given image and compute the representation associated to the attacked image. When using RCDM conditioned on the representation of the adversarial examples, we can visualize if the generated images still belong to the class of the attacked image or not. In Fig. 5 and 30, the adversarial attacks change the dog in the samples to a lion in the supervised setting whereas SSL methods doesn't seem to be impacted by the adversarial perturbations i.e the samples are still dogs until the adversarial attack became visible to the human eye.

## 5.3 Manipulation of representations

Experimental manipulation of representations can be needed to analyze how much specific dimensions of the representation can be associated to specific factors of variation of the data. In a self-supervised setting in which we don't have access to labelled data, it can be difficult to gain insight on how the information about the data is encoded in the representation. We showcase a very simple and heuristic setup to remove the most common information in the representations within a set of the nearest neighbors of a specific example. We experimentally saw that the nearest neighbors of a given representation share often similar factors of variation. Having this information in mind, we investigate how many dimensions are shared in between this set of neighbors. Then, we remove the most common non-zero dimensions by setting them to zero and use RCDM to decode this truncated representation. In Fig. 6, such simple setup induces the removing of all information about the background and the dog to only keep the information about clothing (Only one dog had clothes in the set of neighbors used to find the most common dimensions). Since the information about the dog and the background are removed, RCDM produces images of different clothes. On the third and forth row, instead of setting the most common dimensions to zeros, we set them to the value of other images

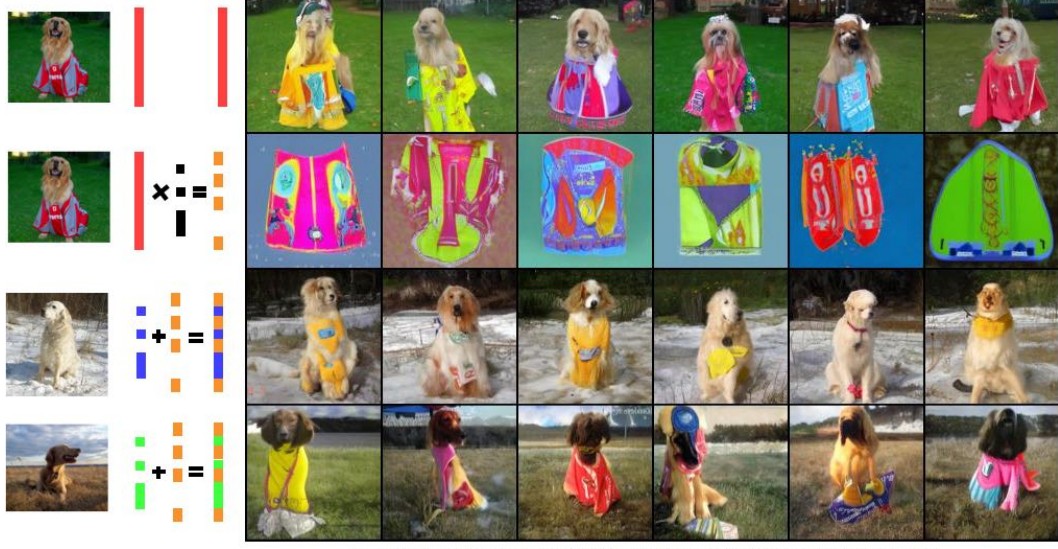

■ zero mask of most common indices where dim of representation is non zero
■ Least common dim of ■ where dim of representation is non zero

Figure 6: Visualization of direct manipulations over the backbone representation space. In this experiment, we find the most common non zero dimension across the neighborhood of the image used as conditioning (top-left dog). On the second row, we set these dimensions to zero and use RCDM to decode the truncated representation. We observe that RCDM produces examples with a variety of clothes meaning that all information about the background and the dog is removed. In the third and forth row, instead of setting the most common dimensions to zero, we set them to the value of the corresponding dimension in the representation associated to the image on the left. As we can see, the corresponding dog get various clothes which were not present in the original image.

at the exact same dimensions. By using these new representations, RCDM is able to generate the corresponding dog with clothes. This setup works better with SSL methods since supervised models learned to put away most of the information that is not needed to predict class labels. We have a similar experiment with background removal in Figure 31.

## 6  CONCLUSION

Most of the Self-Supervised Learning literature uses downstream tasks that require labeled data to measure how good the learned representation is and to quantify its invariance to specific data-augmentations. However one cannot in this way see the entirety of what is retained in a representation, beyond testing for specific invariances known beforehand, or predicting specific labeled factors, for a limited (and costly to acquire) set of labels. Yet, through conditional generation, all the stable information can be revealed and discerned from visual inspection of the samples. We showcased how to use a simple conditional generative model (RCDM) to visualize representations, enabling the visual analysis of what information is contained in a self-supervised representation, without the need of any labelled data. After verifying that our conditional generative model produces high-quality samples (attested qualitatively and by FID scores) and representation-faithful samples, we turned to exploring representations obtained under different frameworks. Our findings clearly separate supervised from SSL models along a variety of aspects: their respective invariances – or lack thereof – to specific image transformations, the discovery of exploitable structure in the representation's dimensions, and their differing sensitivity to adversarial noise.

## 7  REPRODUCIBILITY STATEMENT

The data and images in this paper were only used for the sole purpose of exchanging reproducible research results with the academic community.

Our results should be easily reproducible as:

- RCDM, is based on the same code as Dhariwal & Nichol (2021) (`https://github.com/openai/guided-diffusion`) and uses the same hyper-parameters (See Appendix I of Dhariwal & Nichol (2021) for details about the hyper-parameters).

- To obtain our conditional RCDM, one just needs to replace the GroupNormalization layers in that architecture by a conditional batch normalization layer of Brock et al. (2019) (using the code from `https://github.com/ajbrock/BigGAN-PyTorch`).

- The self-supervised pretrained models we used to extract the conditioning representations were obtained from the model-zoo of VISSL (Goyal et al., 2021) (code from `https://github.com/facebookresearch/vissl`).

- The unconditonal sampling process is straightforward, as explained in Appendix C.

- We are working on cleaning and preparing to release any remaining code glue to easily reproduce the results in this paper.

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

## A MATCHING A REPRESENTATION BY FOLLOWING INPUT GRADIENTS

We want to visualize what kinds of input images would be mapped to the same representation $\boldsymbol{h}$ as that of an input image $\boldsymbol{x}$ by a given network function $f$. This section proposes to sample inputs that fall into that equivalence class by solving an explicit optimization problem, while the next section leverages a conditional diffusion model.

### A.1 PEEKING INTO THE REPRESENTATION-MATCHING INPUT SET

We call the set of inputs that a trained network function $f$ maps to a given representation $\boldsymbol{h}$ the *representation-matching input set*, defined formally as

$$\mathcal{S}(\boldsymbol{h}) \triangleq \{\boldsymbol{x}' \in \mathbb{X} : d(\boldsymbol{h}, f(\boldsymbol{x}')) = 0\}, \tag{1}$$

where $d$ could be any desired distance[4]. We would like to see what kinds of "images" $\mathcal{S}(\boldsymbol{h})$ contains (besides the $\boldsymbol{x}$ that we may have used to obtain representation $\boldsymbol{h}$ to begin with). We tackle this problem through a simple gradient-based optimization. We start from a random input $\boldsymbol{x}^{(0)} \in \mathbb{X}$, sampled from a basic distribution (s.a. uniform). We then performing $T$ gradient steps in input-space towards minimizing objective $d(f(\boldsymbol{x}), \boldsymbol{h})$, i.e. to "match the representation", yielding final sample $\boldsymbol{x}^{(T)}$. Note that the minimizer is usually not unique, so that we can obtain quite different $\boldsymbol{x}^{(T)}$ depending on the random $\boldsymbol{x}^{(0)}$ we started from.

### A.2 DO SAMPLES FROM $\mathcal{S}(f(x))$ LOOK LIKE $x$?

We performed the above-described procedure to sample examples from $\mathcal{S}(f(\boldsymbol{x}))$, using for $f$ the same ResNet50 backbone (He et al., 2016) trained with the DINO SSL criterion (Caron et al., 2021) on ImageNet (Russakovsky et al., 2015). We took $\boldsymbol{x}$ from the validation set of ImageNet. In addition to standard DINO training, we also trained a second SSL network (termed DINO+n) that uses independent additive noise as extra augmentation[5] which led to 71% top-1 Imagenet accuracy. Examples of obtained images are provided in Figure 7. Appendix A.3 has more details and more results for these experiments. We see that even though gradient-based input optimization manages to produce samples that match a target embedding representation, this technique fails to produce realistic images. *Mapping to the same SSL representation as a natural image is not sufficient for being a similar realistic-looking image.*

**The gradient directions are not enough.** The updates producing the sequence $\boldsymbol{x}^{(1)}, \boldsymbol{x}^{(2)}, \ldots, \boldsymbol{x}^{(T)}$ all follow a trajectory that only involves the Jacobian matrix of deep network $f$ at each step. This is due to the chain rule of calculus and reads as

$$\boldsymbol{x}^{(t+1)} = \boldsymbol{x}^{(t)} + \underbrace{\boldsymbol{J}_f(\boldsymbol{x}^{(t)})^T \boldsymbol{u}(\boldsymbol{x}^{(t)})}_{\text{linear combination of Jacobian matrix rows}}, \tag{2}$$

where $\boldsymbol{u}(\boldsymbol{x}^{(t)})$, the linear combination coefficients, is given by $\nabla_{d(f(\boldsymbol{x}),.)}(f(\boldsymbol{x}^{(t)}))$. As a result, it is clear that $\boldsymbol{x}^{(t+1)}$ is constrained to be within the affine space spanned by $\boldsymbol{J}_f(\boldsymbol{x}^{(t)})^T \boldsymbol{u}(\boldsymbol{x}^{(t)})$ and shifted by $\boldsymbol{x}^{(t)}$. Given that $f$ is a mapping from $\mathbb{R}^D$ to $\mathbb{R}^K$ with in general $K < D$, the dimension of that affine space is at most $K$.

We see that the representation and mapping function $f$ obtained through SSL training are by themselves not sufficient to recover corresponding natural-image-like inputs.

### A.3 MORE ON $\mathcal{S}(f(x))$ SAMPLING

In this section, we propose in Fig. 9 additional gradient based matching that employ the projector head of DINO. We also provide in Tab. 1,2, 3 the distances that those gradient based matched input

---

[4]In practice, we may be content with finding elements of a relaxed representation-constrained set $\mathcal{S}_\epsilon(\boldsymbol{h}) \triangleq \{\boldsymbol{x}' \in \mathbb{X} : d(\boldsymbol{h}, f(\boldsymbol{x}')) \leq \epsilon\}$ allowing for a small tolerance $\epsilon$

[5]The motivation was to learn a smoother map $f$ for gradient-based representation matching to be easier. Although the optimization proved no more difficult in the non-noised case, DINO+n yields *qualitatively* markedly different samples, where one can more easily distinguish natural-image like edges. The reason is unclear; one hypothesis is that to reliably discriminate noised instances the representation must focus more on edges.

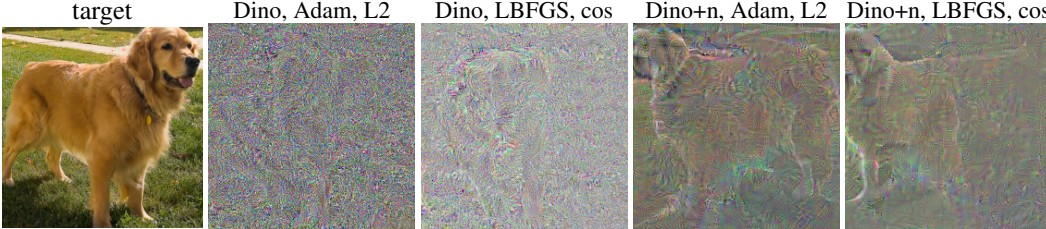

target | Dino, Adam, L2 | Dino, LBFGS, cos | Dino+n, Adam, L2 | Dino+n, LBFGS, cos

Figure 7: Gradient based samples from $\mathcal{S}(\boldsymbol{h})$. Leftmost image $\boldsymbol{x}$ is used to obtain target SSL representation $\boldsymbol{h}$ (2048 dimensions) with either a standard DINO-ResNet50 (Dino) or one trained with additive noise as extra augmentation (Dino+n). A random initialized input is moved so that its representation will match $\boldsymbol{h}$, by minimizing either L2 or cosine distance using Adam or L-BFGS respectively (indicated in column headers). We display the samples $\boldsymbol{x}^{(T)}$ obtained after $T = 10,000$ iterations obtained from the respective optimizers and distances, in all cases starting from a random Gaussian image as $\boldsymbol{x}^{(0)}$. These $\boldsymbol{x}^{(T)}$ have the same SSL representation as $\boldsymbol{x}$ or very close (relative distance $0.4\%, 0.1\%, 3.6\%, 3.3\%$ respectively, see details in appendix Tab. 1,2,3), but do not resemble natural images. Samples obtained from Dino+n look slightly more natural: we distinguish faint edges similarly shaped to the original. Similar experiments but applied on the lower-dimensional projection head embeding are reported in Figure 9 in appendix.

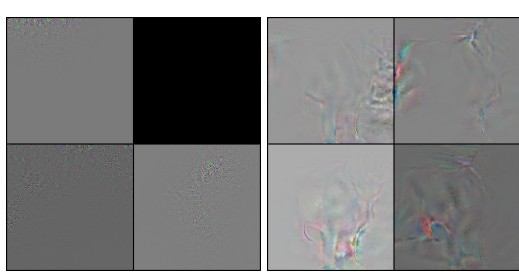

Figure 8: Depiction of four rows of the Jacobian matrix $\boldsymbol{J}_f(\boldsymbol{x})$ for the input $\boldsymbol{x}$ given in Figure 7, with $f$ being a Resnet50 trained with standard DINO (**left**) and additive noise (DINO+n) (**right**). The Jacobian matrix in the noisy case shows more structures looking somewhat more natural-image like. Recalling (2), this observation justifies the more natural images observed in Figure 7. Additional rows provided in the appendix in Figure 10)

| distance | $\ell_2$ | $\ell_1$ | cosine | relative $\ell_2$ (%) | relative $\ell_1$ (%) | relative cosine (%) |
|---|---|---|---|---|---|---|
| Adam plateau | 0.8 | 30.0 | 0.0 | 0.1 | 5.5 | 0.1 |
| Adam cosine | 0.4 | 11.0 | 0.0 | 0.1 | 2.1 | 0.1 |
| GD plateau | 2.8 | 48.0 | 0.1 | 0.4 | 8.8 | 5.7 |
| GD cosine | 2.2 | 17.0 | 0.1 | 0.3 | 3.1 | 7.8 |
| L-BFGS plateau | 0.1 | 23.0 | 0.0 | 0.0 | 4.3 | 0.0 |
| L-BFGS cosine | 0.1 | 26.0 | 0.0 | 0.0 | 4.9 | 0.0 |

Table 1: We depict here the final value of the input optimization step ($\boldsymbol{x}^{(T)}$). We experiment with different distances (each column) and we provide the actual value of the distance along with a relative distance which is obtained by $100 - 100 \times |d(f(\boldsymbol{x}), f(\boldsymbol{x}^{(0)})) - d(f(\boldsymbol{x}), f(\boldsymbol{x}^{(T)}))|/d(f(\boldsymbol{x}), f(\boldsymbol{x}^{(0)}))$. That is, the relative distance gives a proportion of how close to the target is the obtained representation as a ratio with respect to the distance using the initial (random) image, value between 0 and 100. In this table, we are looking at the DINO model. We provide the noise models in the below tables.

can read in term of representation from a target one. In the next section we also provide additional theoretical arguments supporting the challenge of following gradient directions to obtain realistic samples from $\mathcal{S}$.

## A.4   DEEP NETWORKS AND CPAS

In this section we propose to further characterize what $\mathcal{S}(\boldsymbol{h})$ looks like by using a specific form for the DN input output mapping.

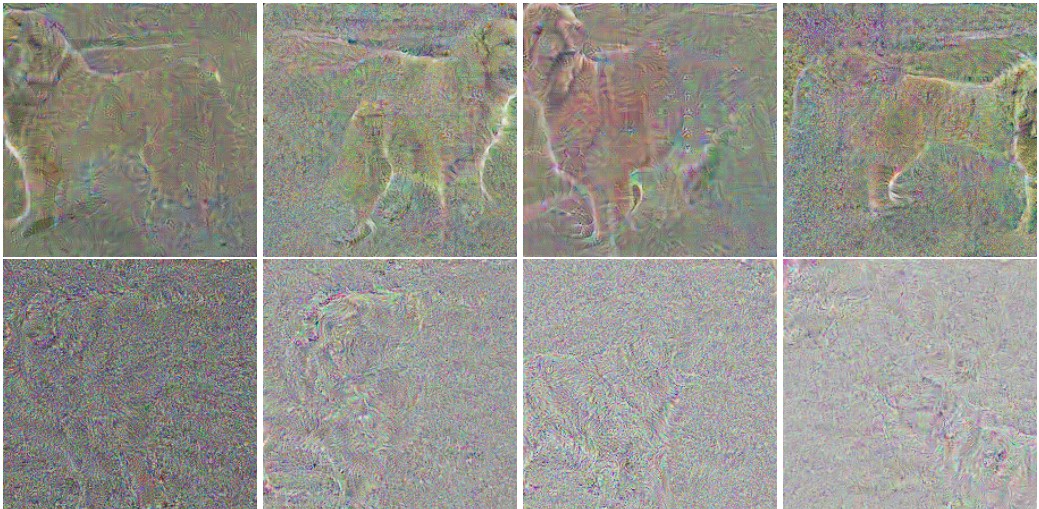

Figure 9: Reprise of Fig. 8 but now when considering the mapping to be the resnet50 backbone and the projection head of DINO. Top row is when using DINO+noise and the bottom row is when using standard DINO. As was the case when using the resnet50 backbone only, we do not obtain realistic inputs from $\mathcal{S}(\boldsymbol{x})$ when following gradient directions.

| distance | $\ell_2$ | $\ell_1$ | cosine | relative $\ell_2$ (%) | relative $\ell_1$ (%) | relative cosine (%) |
|---|---|---|---|---|---|---|
| Adam plateau | 3.1 | 54.0 | 0.0 | 0.9 | 13.5 | 0.9 |
| Adam cosine | 3.6 | 41.0 | 0.0 | 1.0 | 10.2 | 0.8 |
| GD plateau | 12.2 | 107.0 | 1.3 | 3.5 | 26.6 | 91.8 |
| GD cosine | 14.0 | 90.0 | 1.3 | 4.1 | 22.5 | 95.1 |
| L-BFGS plateau | 2.2 | 50.3 | 0.0 | 0.6 | 12.4 | 0.7 |
| L-BFGS cosine | 3.3 | 560.0 | 0.0 | 0.9 | 61.6 | 0.5 |

Table 2: Reprise of Tab. 1 but with DINO noise

| distance | $\ell_2$ | $\ell_1$ | cosine | relative $\ell_2$ (%) | relative $\ell_1$ (%) | relative cosine (%) |
|---|---|---|---|---|---|---|
| Adam plateau | 1.8 | 52.4 | 0.0 | 0.3 | 15.6 | 0.3 |
| Adam cosine | 2.8 | 39.3 | 0.0 | 0.5 | 11.7 | 0.5 |
| GD plateau | 27.2 | 131.0 | 0.7 | 4.5 | 39.3 | 85.4 |
| GD cosine | 50.1 | 170.0 | 0.7 | 8.4 | 50.9 | 89.9 |
| L-BFGS plateau | 3.6 | 53.7 | 0.0 | 0.6 | 16.0 | 0.5 |
| L-BFGS cosine | 3.4 | 52.3 | 0.0 | 0.6 | 15.6 | 0.5 |

Table 3: Reprise of Tab. 1 but with DINO noise ++

Without loss of generality we consider a mapping $f$ that is continuous piecewise affine (CPA), as is the case for most DNs(Balestriero & Baraniuk, 2018). The DN input-output is then given by

$$f(\boldsymbol{x}) = \sum_{\omega \in \Omega} (\boldsymbol{A}_\omega \boldsymbol{x} + \boldsymbol{b}_\omega) \, 1_{\{\boldsymbol{x} \in \omega\}}, \qquad (3)$$

with $\Omega$ a partition of the DN input space. In the case of $f$ being smooth, a simple approximation argument will allow to fall back to the above setting (Daubechies et al., 2021). Using this formulation, we can now characterize more precisely the form of $\mathcal{S}(f(\boldsymbol{x}))$ as follows

$$\mathcal{S}(f(\boldsymbol{x})) = \cup_{\boldsymbol{x}' \in \mathcal{X}(\boldsymbol{x})} \Big( \omega(\boldsymbol{x}') \cap \underbrace{\{\boldsymbol{x}' + \boldsymbol{u}, \boldsymbol{u} \in \ker\left(\boldsymbol{A}_{\omega(\boldsymbol{x}')}\right)\}}_{\text{linear subspace } \ker(\boldsymbol{A}_{\omega(\boldsymbol{x}')}) \text{ shifted by } \boldsymbol{x}'} \Big), \qquad (4)$$

where $\omega(\boldsymbol{x}')$ is the region from $\Omega$ in which $\boldsymbol{x}'$ lives in, and $\mathcal{X}(\boldsymbol{x})$ is a finite set of inputs that depend on $\boldsymbol{x}$ such that each point lives in a separate region from the others as in $\forall \boldsymbol{u}, \boldsymbol{v} \in \mathcal{S}(f(\boldsymbol{x}))^2, \omega(\boldsymbol{u}) = \omega(\boldsymbol{v}) \iff \boldsymbol{u} = \boldsymbol{v}$.

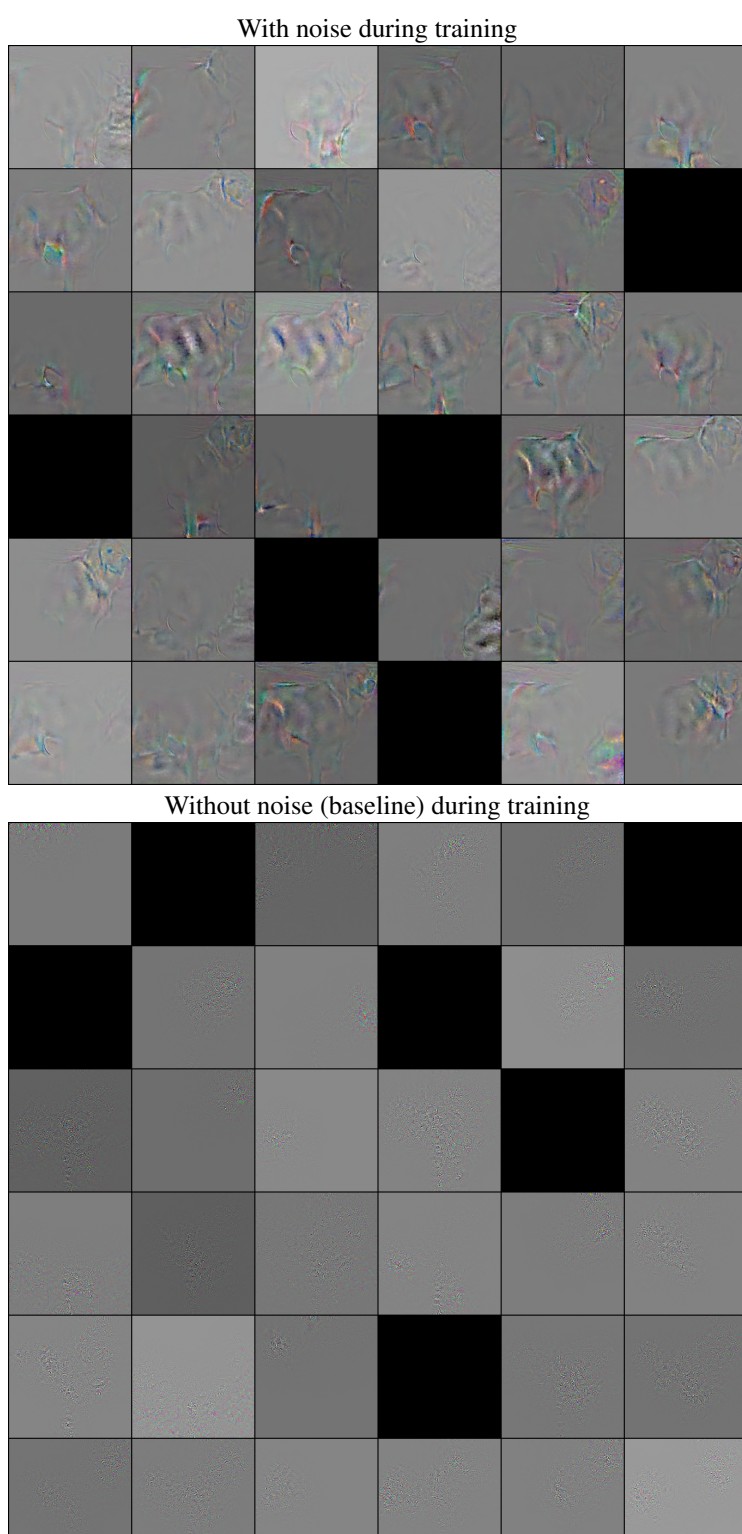

Figure 10: Depiction of 36 rows of the Jacobian matrix of a trained DINO model that either employed Gaussian noise on the images during training (top) or did not (bottom). Clearly the use of noise during training produce Jacobian matrices with more natural patterns.

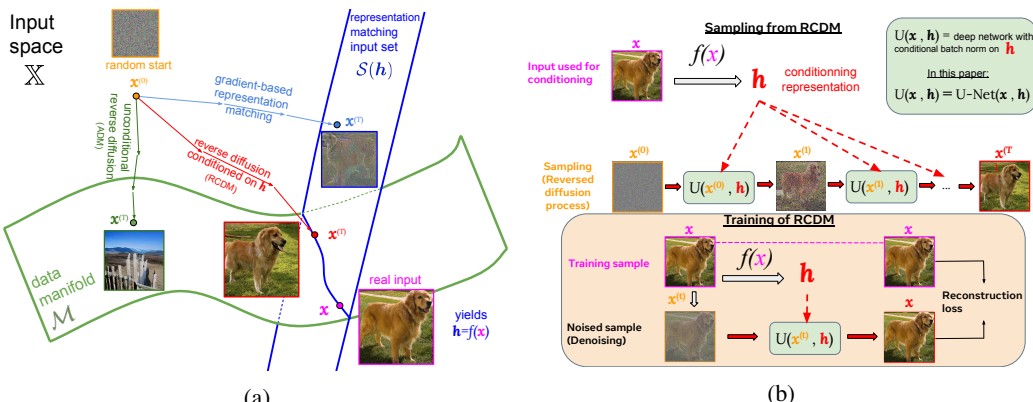

Figure 11: (a) Illustration of considered image generation methods. A real input $x$ yields representation $h$. All methods start from a random noise image $x^{(0)}$. Gradient-based representation matching (light blue arrows) will move it towards $\mathcal{S}(h)$ i.e. until its representation matches $h$, but won't land on the data-manifold $\mathcal{M}$. Unconditional reverse diffusion (ADM model, green arrows) will move it towards the data manifold. Our representation-conditioned diffusion model (RCDM, red arrows) will move it towards $\mathcal{M} \cap \mathcal{S}(h)$, yielding a different natural-looking image with the same given representation. (b) Representation-Conditionned Diffusion Model (RCDM). From a diffusion process that progressively corrupts an image, the model learns the reverse process by predicting the noise that it should remove at each step. We also add as conditioning a vector $h$, which is the representation given by a SSL or supervised model for a given image $c$. Thus, the network is trained explicitly to denoise towards a specific example given the corresponding conditioning. The diffusion model used is the same as the one presented by Dhariwal & Nichol (2021) with the exception of the conditioning on the representations.

**The optimization problem needs to be constrained.** The first challenge of our method comes from the fact that the set $\mathcal{S}(f(x))$ consists of a union of affine subspace that are highly localized in the input space (recall (4)). In fact, the regions $\omega \in \Omega$ are often extremely localized in the input space, especially as the architecture involves many layers (Montúfar et al., 2014; Balestriero et al., 2019). In addition to that optimization difficulty, we have that the equivalence class does not constrain the inputs to lie within the data manifold. In fact, each affine subspace that form $\mathcal{S}(f(x))$ is very high dimension as we emphasize below.

**Proposition 1.** *Given a model $f : \mathbb{X} \mapsto \mathbb{H}$, the set $\mathcal{S}(f(x))$ is a union of $\mathrm{Card}(\mathcal{X}(x))$ affine subspaces, each with dimension at least $D - K$.*

In other words, regardless of the chosen distance $d$, as soon as the dimensions of the affine subspaces forming $\mathcal{S}(f(x))$ are greater than the dimension of the data manifold $\mathcal{M}$ (a sufficient condition being $D - S > \dim(\mathcal{M})$) we obtain that $\mathcal{S}(f(x))$ contains samples that do not belong to $\mathcal{M}$. That is, performing gradient descent from randomly initialized samples $x^{(0)}$ will almost surely produce samples $x^{(T)} \notin \mathcal{M}$. This is particularly true for a case such as Imagenet in which $D = 150528$ and $S = 2048$.

## B  CONDITIONAL AND SUPER-RESOLUTION SAMPLING WITH RCDM

As presented in the main text, we introduce RCDM to generate samples that preserved well the semantics of the images used for the conditioning. As showed in Figure 11a, RCDM is constraint to map back the representation to the manifold of real images which answers the concerns raised in Appendix A. The training of the model is very simple and presented in Figure 11b. We show in Figure 12 additional samples of RCDM when conditioning on the SSL representation of ImageNet validation set images (which were never used for training). We observe that the information hidden in the SSL representation is so rich that RCDM is almost able to reconstruct entirely the image used for conditioning. To further evaluate the abilities of this model, we present in Figure 13 a similar experiment except that we use out of distribution images as conditioning. We used cell images from microscope and a photo of a status (Both from Wikimedia Commons), sketch and cartoons from

PACS (Li et al., 2017), image segmentation from Cityscape (Cordts et al., 2016) and an image of the Earth by NASA. Even in the OOD scenario, RCDM is able to generate images that are very close to the one used as conditioning because of the richness of ssl representations.

We also use the super-resolution model presented by Dhariwal & Nichol (2021) to generate images of higher resolutions. In Figure 14, we use the small images on the top of the bigger images as conditioning for a RCDM trained on images of size 128x128. Then, we feed the 128x128 samples into the super-resolution model of Dhariwal & Nichol (2021) to get images of size 512x512. Since the model of Dhariwal & Nichol (2021) is conditional and need labels, we used a random label when upsampling from RCDM. Despite using the "wrong" label, the high resolution samples are still very close to the conditioning. This show that RCDM can be used jointly with a super-resolution model to sample high fidelity images in the close neighborhood of the conditioning.

To verify how well our model can produces realistic samples from different combinations of representations, we take two images from which we compute their representations and perform a linear interpolation between those. This give us new vectors of representation that can be used as conditioning for RCDM. We can see on Figure 15 and Figure 16 that RCDM is able to generate samples that contains the semantic characteristics of both images.

Finally, in Figure 17, we search the nearest neighbors of a series of samples in the ImageNet training set. As demonstrated by Figure 17, RCDM samples images that are new and far enough from images belonging to the training set of ImageNet.

## C   A HIERARCHICAL DIFFUSION MODEL FOR UNCONDITIONAL GENERATION

We provided a novel and conditional generative model based on a given latent representation e.g. from a SSL embedding, and a diffusion model. This provided tremendous insights into interpreting what is encoded in those representations. We can go one step further and augment this conditional model with an unconditional one that can generate those representations. This will provide us with the ability to generate new samples without the need to condition on a given input. As a by-product, it will allow us to quantify the quality of our generative process in an unconditional manner to fairly compare against state-of-the-art generative models.

We shall recall that our goal is to employ the conditional generative model to provide understanding into learned (SSL) representations. The unconditional model is only developed to compare our generative model and ensure that its quality is reliable for any further down analysis. As such, we propose to learn the representation distribution in a very simple manner via the usual Kernel Density Estimation (KDE). That is, the distribution is modeled as

$$p(\boldsymbol{h}) = \frac{1}{N} \sum_{n=1}^{N} \mathcal{N}(\boldsymbol{h}; f(\boldsymbol{x}_n); I\sigma)$$

with $\sigma$ set to $0.01$. By using the above distribution, we are able to sample representations $\boldsymbol{h}$ to then sample images $\boldsymbol{x}$ conditionally to that $\boldsymbol{h}$ using our diffusion model. We provide some samples in Figure 21 to show that even with our very simple conditioning, our method is still able to generate realistic images.

## D   ON THE CLOSENESS OF THE SAMPLES IN THE REPRESENTATION SPACE

Even if we show that RCDM is able to generate images that seems visually close to the image used for the conditioning, it's still unclear how close those images are in the representation space. We can compute euclidean distances but to know how close the generated samples are to the conditioning, we need to have references that can be used to compare this distance with. As references, we compute the euclidean distance between a conditioning image and random images in the validation set of ImageNet, random images belonging to the same class as the conditioning, the closest images in the training set, the conditioning image on which we applied single data augmentations and the conditoning image on which we applied the data augmentation performed by Swav and Dino (Caron et al., 2020; 2021). The results can be seen in Figure 22 for a RCDM trained with Dino representations and in Figure 23 for a RCDM trained with SimCLR representations. On both Figure, we observe that

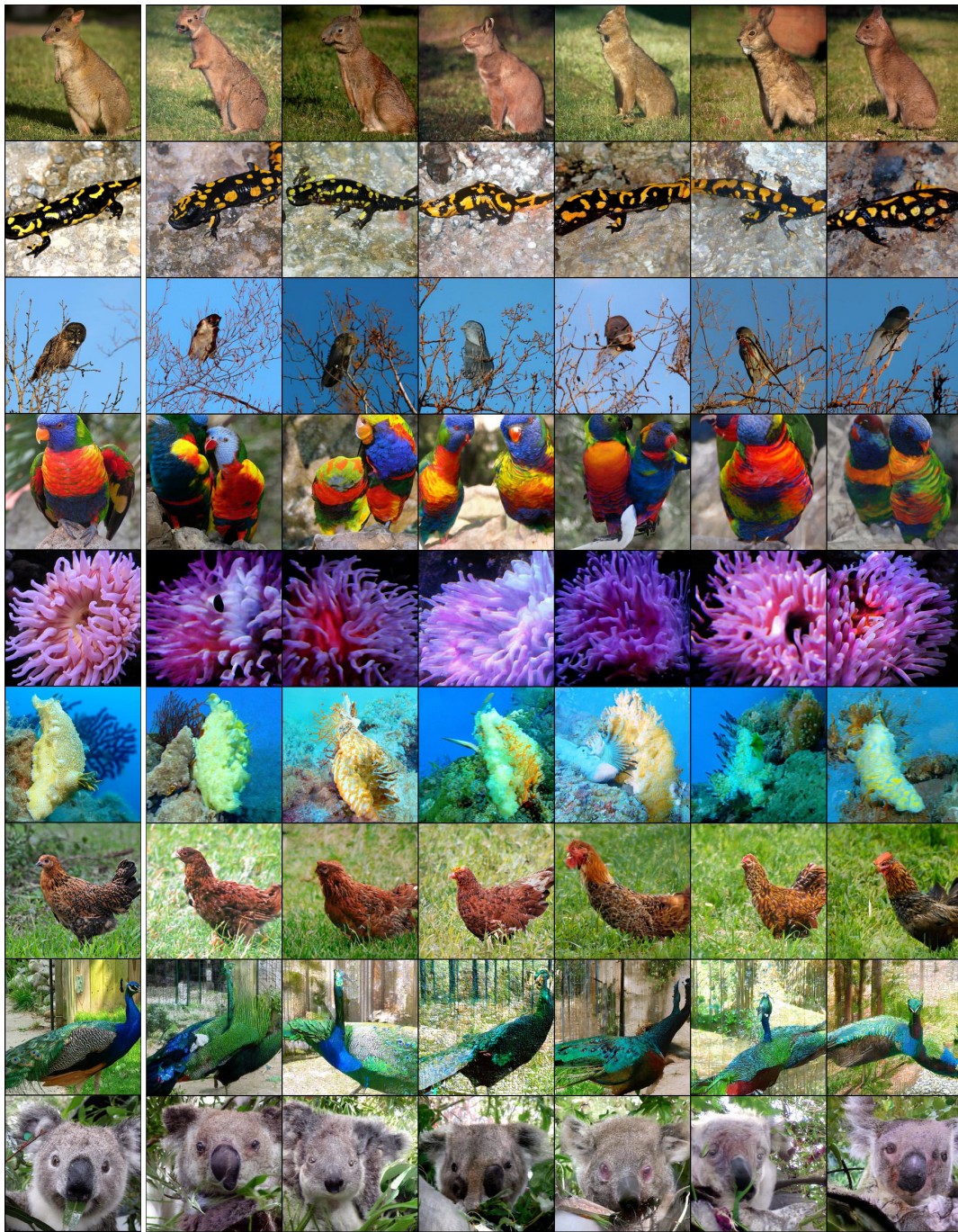

Figure 12: Generated samples from RCDM on 256x256 images trained with representations produced by Dino. We put on the first column the images that are used to compute the representation conditioning. On the following column, we can see the samples generated by RCDM. It is worth to denote our generated samples are qualitatively close to the original image.

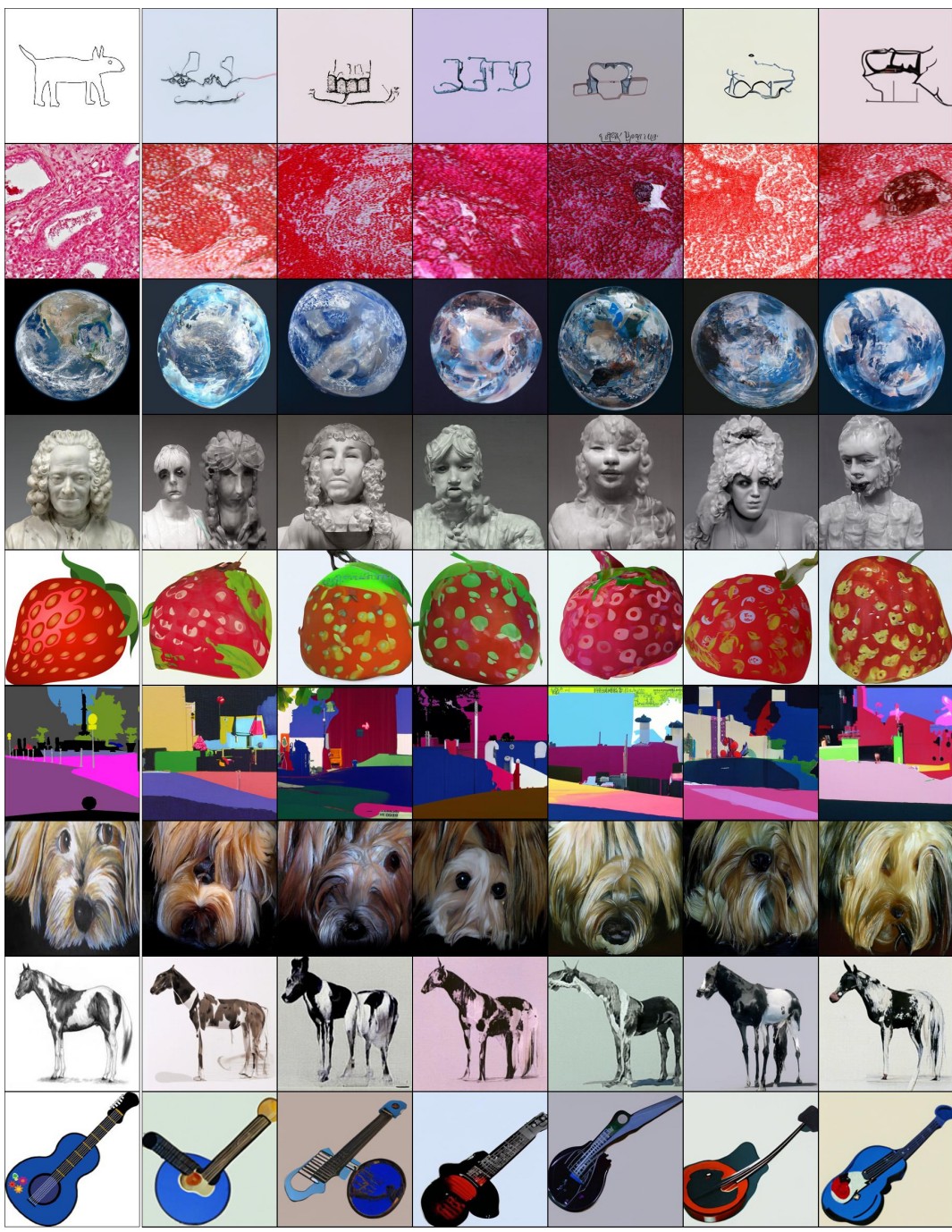

Figure 13: Generated samples from RCDM model on 256x256 images trained with representations produced by Dino on Out of Distribution data. We put on the first column the images that are used to compute the representation. On the following column, we can see the samples generated by RCDM. It is worth to denote our generated sample are close to the original image. The images used for the conditioning are from Wikimedia Commons, Cityscapes (Cordts et al., 2016), PACS (Li et al., 2017) and the image of earth from NASA.

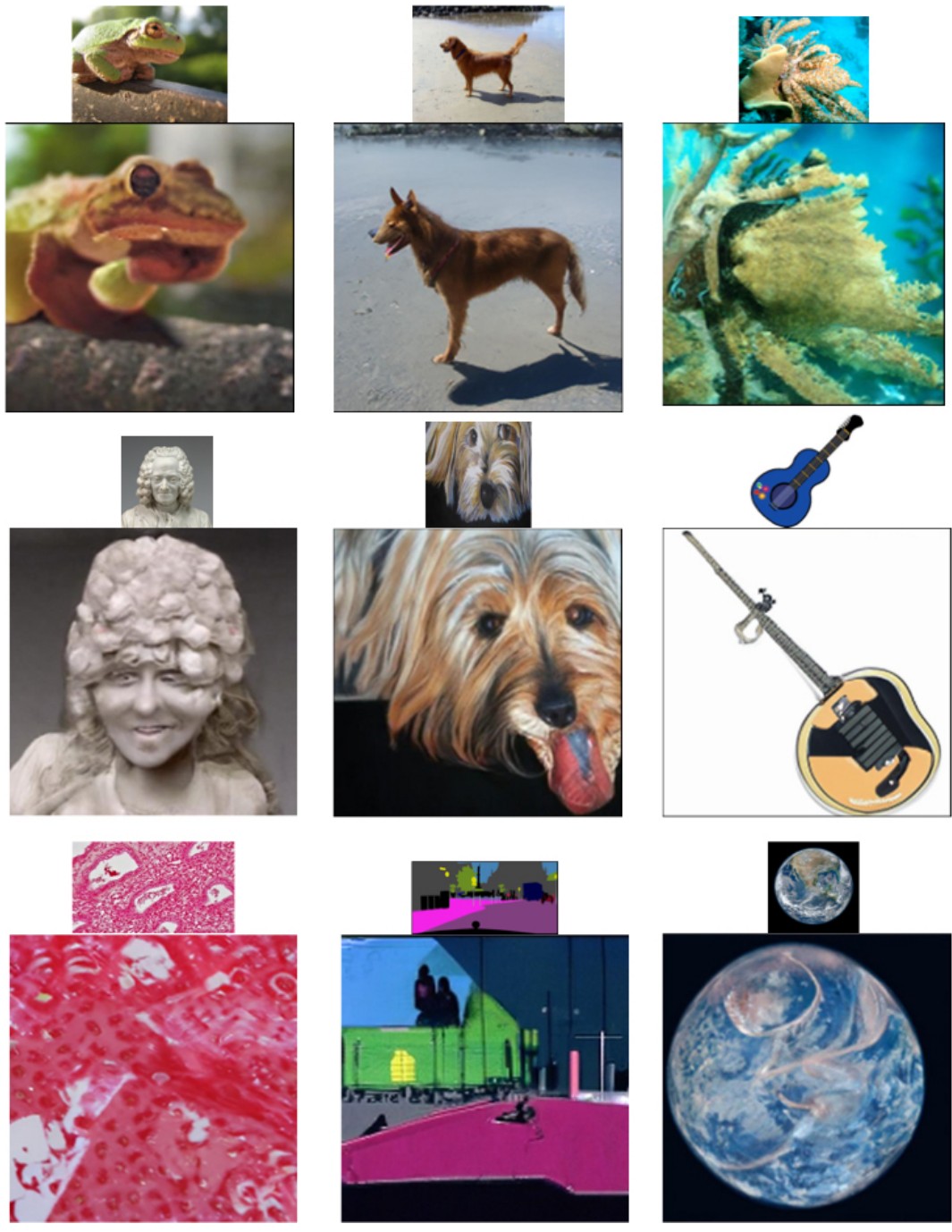

Figure 14: High resolution samples from our conditional diffusion generative model using the super resolution model of Dhariwal & Nichol (2021). We use the small images on the top of each bigger image as conditioning for a diffusion model trained with Dino representation on 128x128 images. Then, we feed the samples generated to the super resolution model of Dhariwal & Nichol (2021) which produces images of size 512x512. Since the super resolution model is conditional, we sample a random label. We note that the high resolution samples are still very close to the conditioning.

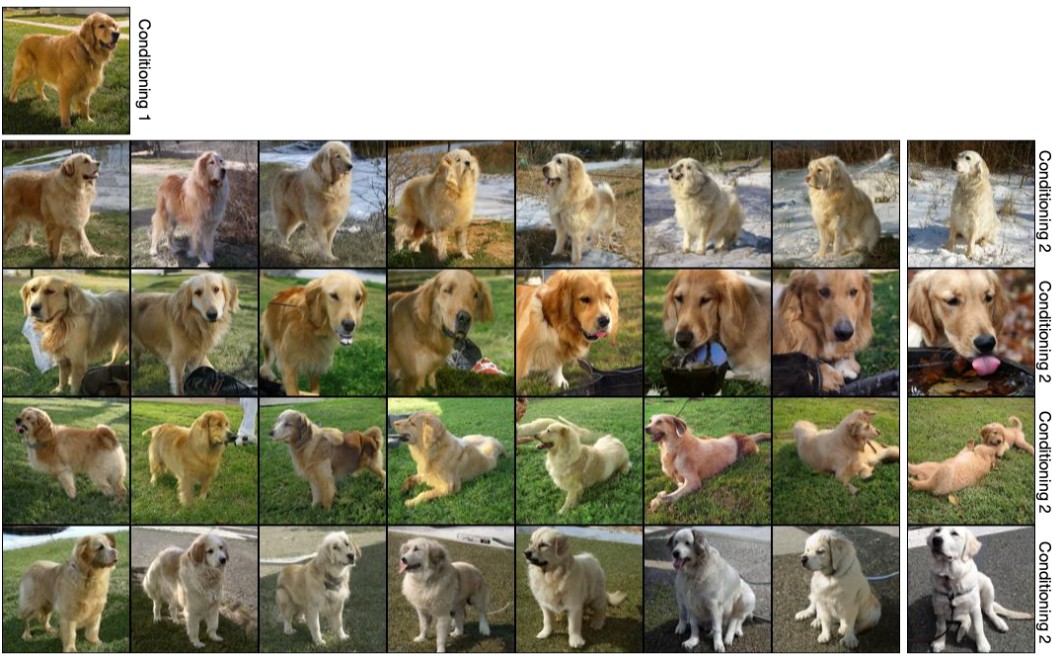

(a) Linear interpolation between the image of the golden retriever in conditioning 1 with various other images belonging to the same class as conditioning 2.

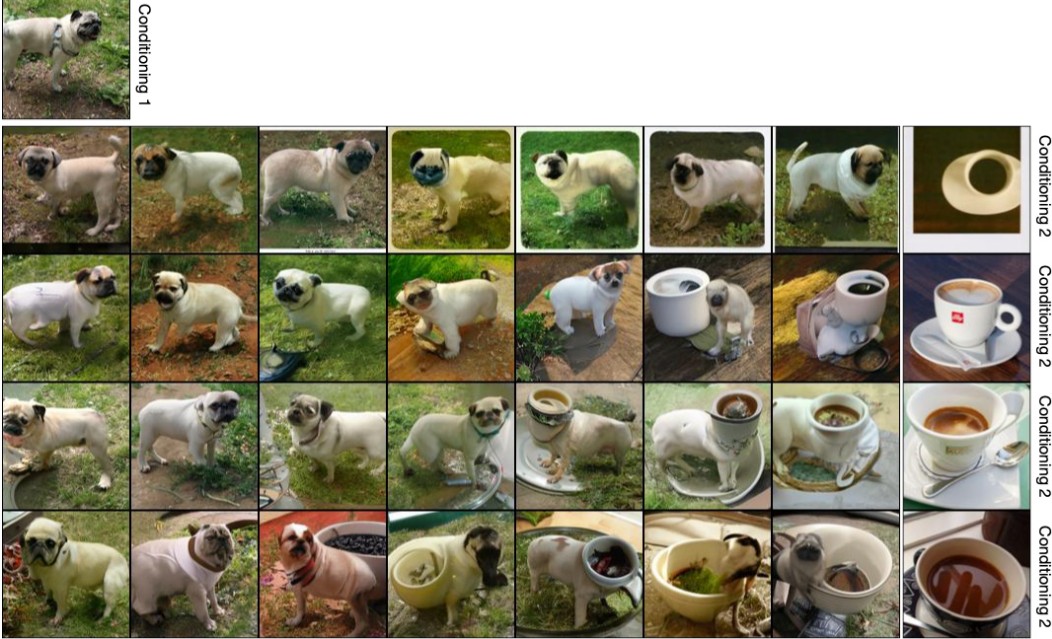

(b) Linear interpolation between the image of the pug in conditioning 1 with various other images belonging to the espresso class as conditioning 2.

Figure 15: Each vectors that result from the linear interpolation is feed to a RCDM trained with Barlow Twins representation.

the generated images with RCDM are closer to the conditioning than the closest neighbors in the entire training set of ImageNet. We also computed the mean and reciprocal mean rank in the main paper (Table 2b) which show that for most SSL models the closest examples in the representation space of the generated images is the image used as conditioning. We also added Figure 18 to show

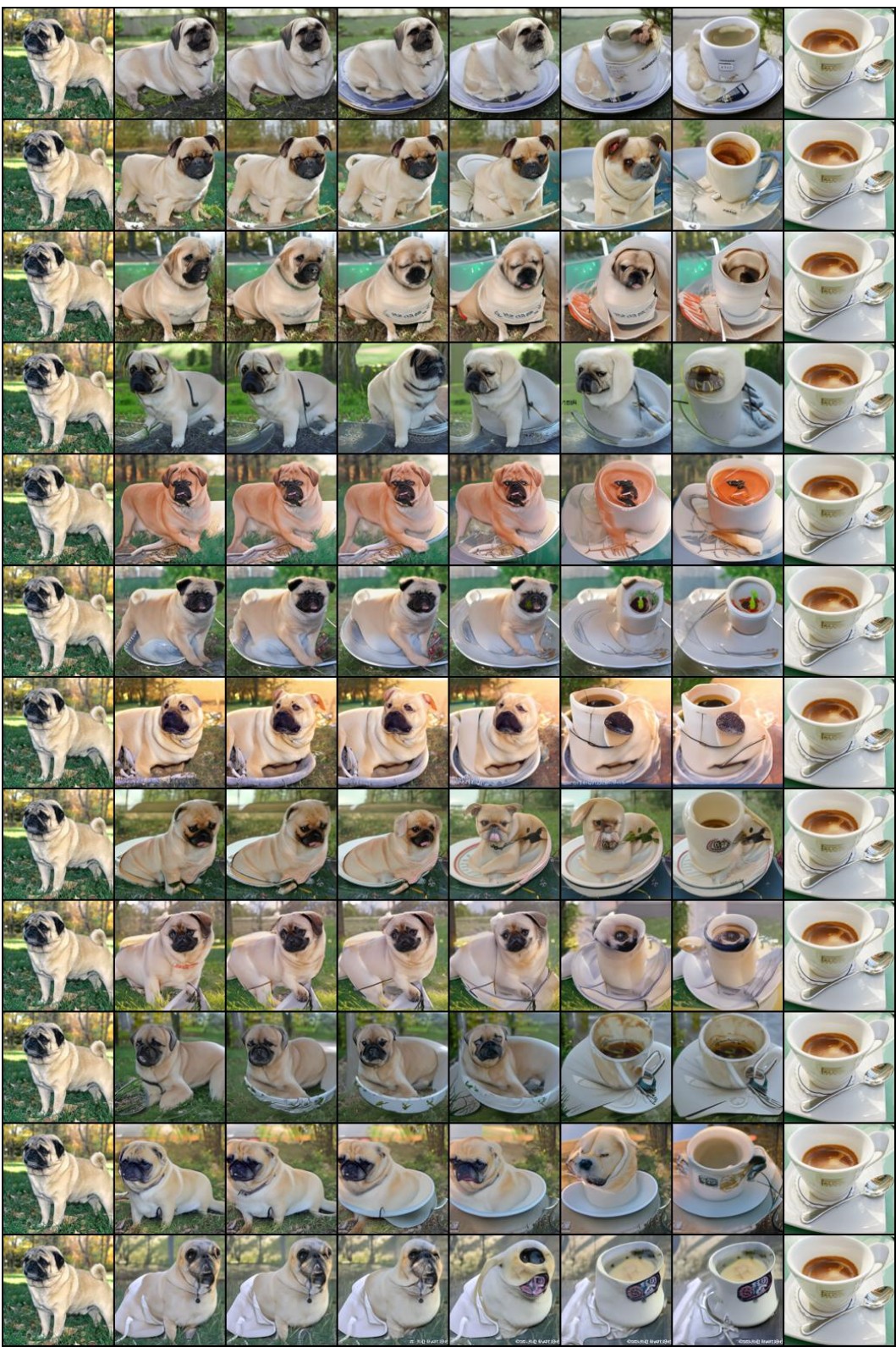

Figure 16: Diversity of the samples generated by RCDM on interpolated representations. Each row corresponds to different random noise for the same conditioning. On the first and last column are the real images used for the interpolation. All of the images in-between those rows are samples from RCDM.

which rank is associated to samples generated by RCDM. For SimCLR, the rank is mostly always 1 whereas we got more diversity for the supervised case. This difficulty of RCDM to generated samples which have their representation that map back to the one used for the conditioning can be explain by the nature of a supervised training. In such scenario, the encoder is trained to map a big set of images (often a specific class) to a specific type of representation whereas SSL models are explicitly train to push each examples farther away from each others. Thus, it seems more likely that a little perturbation on the supervised representation induces a change of nearest neighbor. This hypothesis is supported by Figure 30 which show that small adversarial attack are enough to induces a change of class in the representation which is not the case for SSL encoders.

## E  ANALYSIS OF REPRESENTATIONS LEARNED WITH SELF-SUPERVISED MODEL

Having generated samples that are close in the representation space to a conditioning image can give us an insight on what's hidden in the representations learned with self-supervised models. As demonstrated in the previous section, the samples that are generated with RCDM are really close visually to the image used as conditioning. This give an important proof of how much is kept inside a SSL representation. However, it's also important to consider how much this amount of "hidden" information varied depending on the SSL representation that is used. Therefore, we train several RCDM on SSL representations given by VicReg (Bardes et al., 2021), Dino (Caron et al., 2021), Barlow Twins (Zbontar et al., 2021) and SimCLR (Chen et al., 2020). In many applications that used self-supervised models, the representation that is used is the one corresponding to the backbone of the ResNet50. Usually, the representation given by the projector of the SSL-model (on which the SSL criterion is applied) is discarded because the results on many downstream tasks like classification is not as good as the backbone. However, since our work is to visualize and better understand the differences between SSL representations, we also trained RCDM on the representation given by the projector of Dino, Barlow Twins and SimCLR. In Figure 25 and 26 we condition all the RCDM with the image labelled as conditioning and sample 9 images for each model. We observe that Dino representation does not allow much variance meaning that even information about the pose of the animal is kept inside the representation. In contrast, the SimCLR representation seems to be more invariant to the pose of the kangaroo, maybe too much because on many images the kangaroo become a rabbit. VicReg seems to be more robust in the sense that the animal doesn't cross the class boundary despite changes in the background. If we look at the projector of the SSL models, the generated samples have a higher variance except for Barlow Twins. This can be explained by the fact that the dimension of the representation given by the projector has a size of 8192 which is much bigger than the one used by other methods.

To further compare and analyse the different SSL models, we visualize how much SSL representations can be invariant with respect to a transformation that is applied on the conditioning image. In Figure 27, we apply several Data Augmentation: Vertical shift, Zoom out, Zoom In, Grayscale and a Collor Jitter on a given conditioning image. Then we compute the SSL representations of the transformed image with different SSL models and use our corresponding RCDM to see how much the samples have changed with respect to the samples generated on the vanilla conditioning image. We observe that the representation (the 2048 backbone one) of all SSL methods are not invariant to scale and change of colors. Whereas the representation of the projector doesn't seem to take into account any small transformation in the original conditioning outside the scale for Dino. For SimCLR, there is still some information about the background that is kept in the representation however the samples are not as close visually with respect to the 2048 representation. Barlow Twins is interesting because there isn't much differences between the backbone representation (2048) one and the representation of the projector (Size 8192). With the exception that this last representation seems to be more invariant to color shift than the backbone one.

### E.1  VISUALIZATION OF ADVERSRIAL EXAMPLES

We use RCDM to visualize adversarial examples for different models. For each model, we trained a linear classifier on top of their representations to predict class labels for the ImageNet dataset. Then, we use FGSM attacks over the trained model using a NLL loss to generate adversarial examples. In Figure 30 we show the adversarial examples that are created for each model, the samples generated

by RCDM with respect to the representation of the adversarial perturbed example and the class label predicted by the linear classifier over the adversarial examples. The supervised model is very sensitive to the attack whereas SSL models seems more robust.

### E.2 MANIPULATION OF SSL REPRESENTATIONS

It is also possible to manipulate SSL representations to generate new images. We try to apply addition and subtractions over SSL representations (similarly to what has been done in NLP). From two different images, we compute the difference between the two corresponding representations and add the difference vector to a third image. Figure 33 shows that it is possible to apply such transformations meaningfully in the SSL space. We also used another setup where we choose specific dimensions in the representation based on how many times these dimensions are non zero in the representation space of a set of neighbors. Then we set this dimension to zero which surprisingly induces the removing of the background in the generated images. We also replace them by the same corresponding dimension of another images which induces a change of background toward the one of the new image. Results are shown in Figure 31.

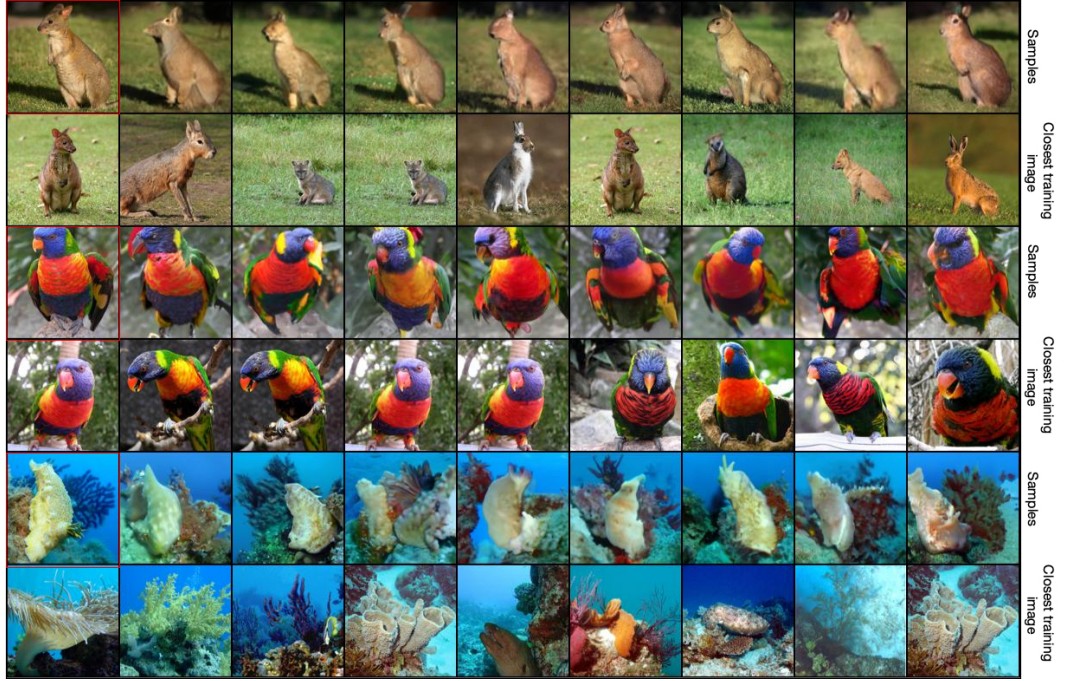

(a) Closest real images (ImageNet training set) from images sampled with RCDM trained on Dino (backbone) representation (2048).

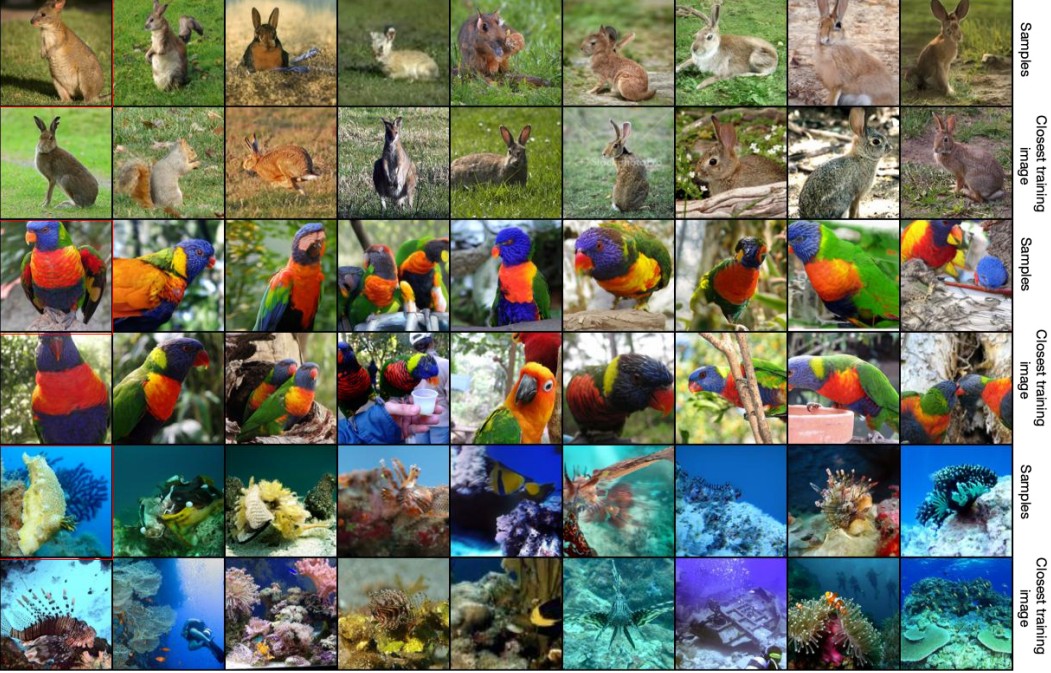

(b) Closest real images (ImageNet training set) from images sampled with RCDM trained on Dino projector representation (256).

Figure 17: We find the nearest neighbors in the representation space of samples generated by RCDM. The images in the red squared are the ones used for conditioning.

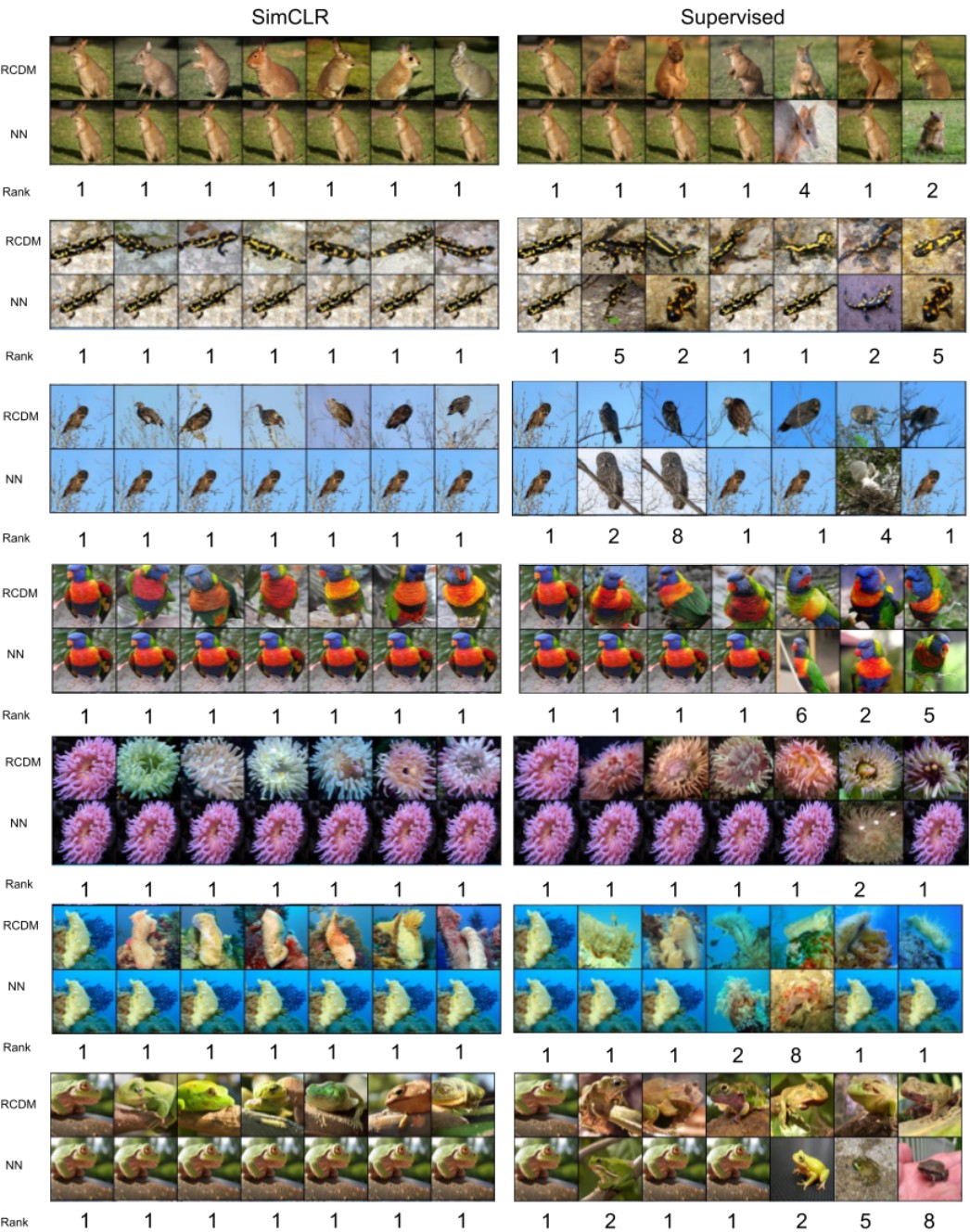

Figure 18: After generating samples with respect to a specific conditioning, we compute back the representation of the generated samples and find which are the closest neighbors in the validation set. Then, we compute the rank of the original image that was used as conditioning within the set of neighbors. When the rank is one, it implies that the nearest neighbors of the generated samples is the conditioning itself, meaning that the generated samples have their representation that is very close in the representation space to the one used as conditioning. We can see that for SimCLR, the generated samples are much closer in the representation space to their conditioning than the supervised representation. This is easily explain by the fact that supervised model learn to map images from a same class toward a similar representation whereas SSL models try to push further away different examples.

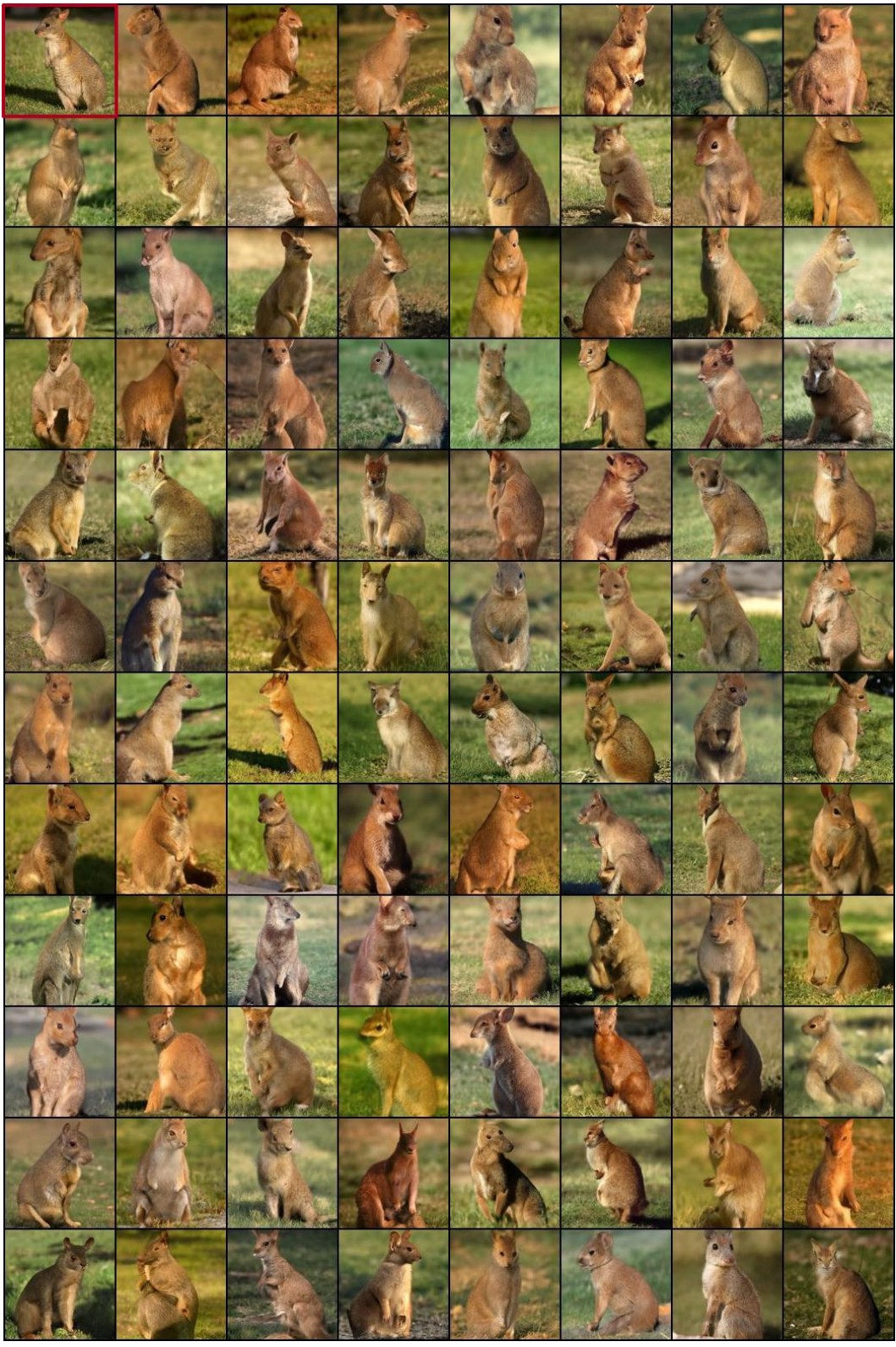

Figure 19: Visual analysis of the variance of the generated samples for a specific image when using a supervised encoder. The first image (in red) in the one used as conditioning.

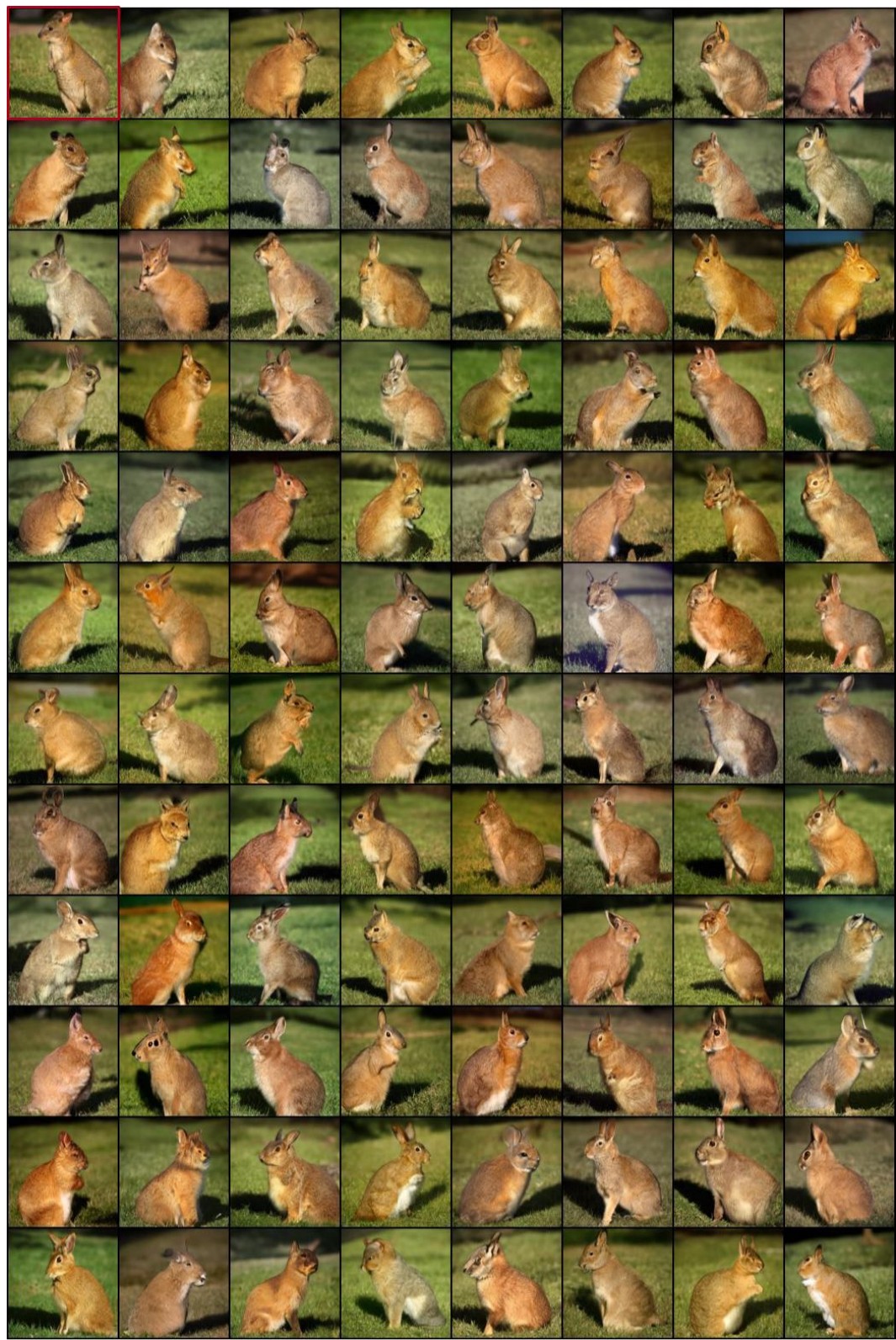

Figure 20: Visual analysis of the variance of the generated samples for a specific image when using a SimCLR encoder. The first image (in red) in the one used as conditioning.

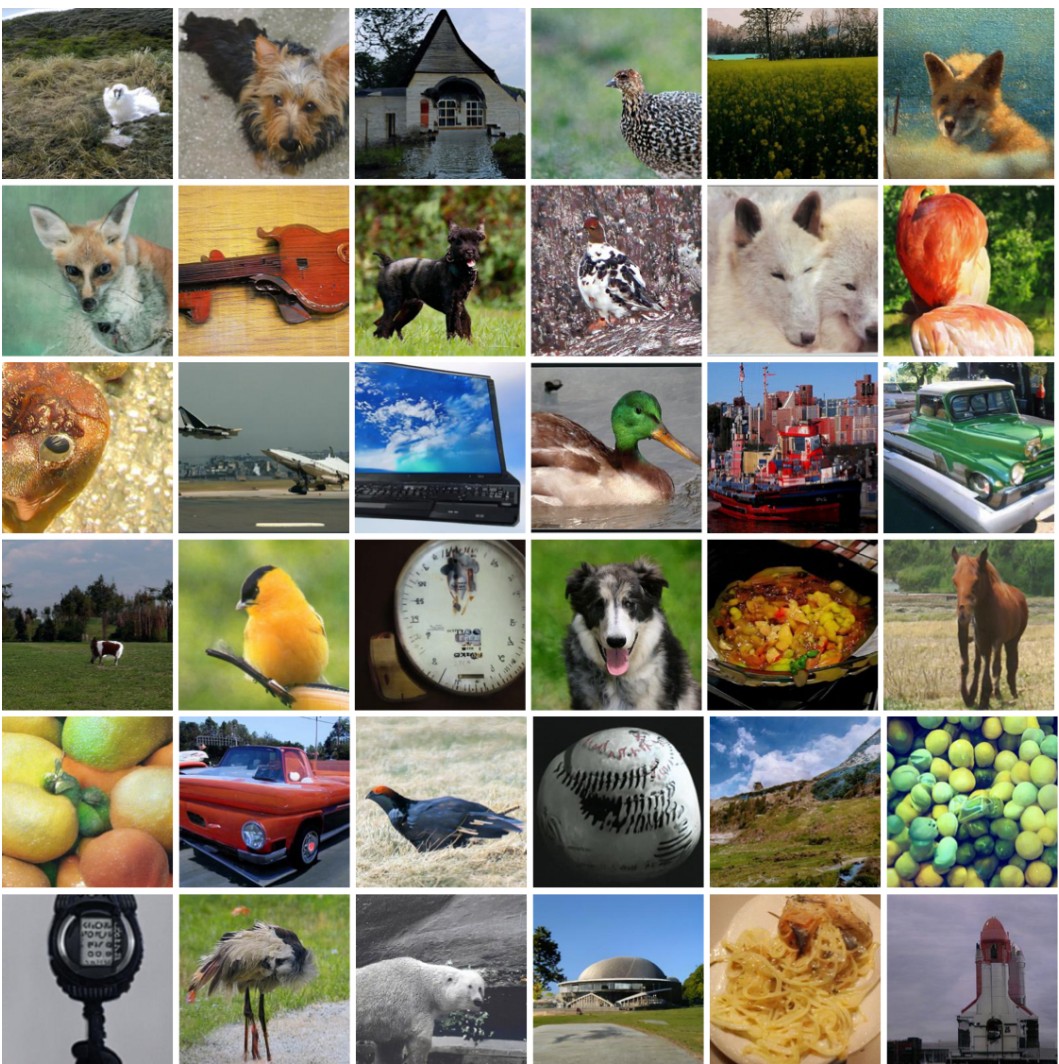

Figure 21: Unconditional generation following the protocol of section C. Our simple generative model of representations consists in applying a small Gaussian noise over representation computed from random training images of ImageNet. We use these noisy vector as conditioning for our 256x256 RCDM trained with Dino representations. We note that the generated images looks realistic despite some generative artefact like a two-headed dog and an elephant-horse.

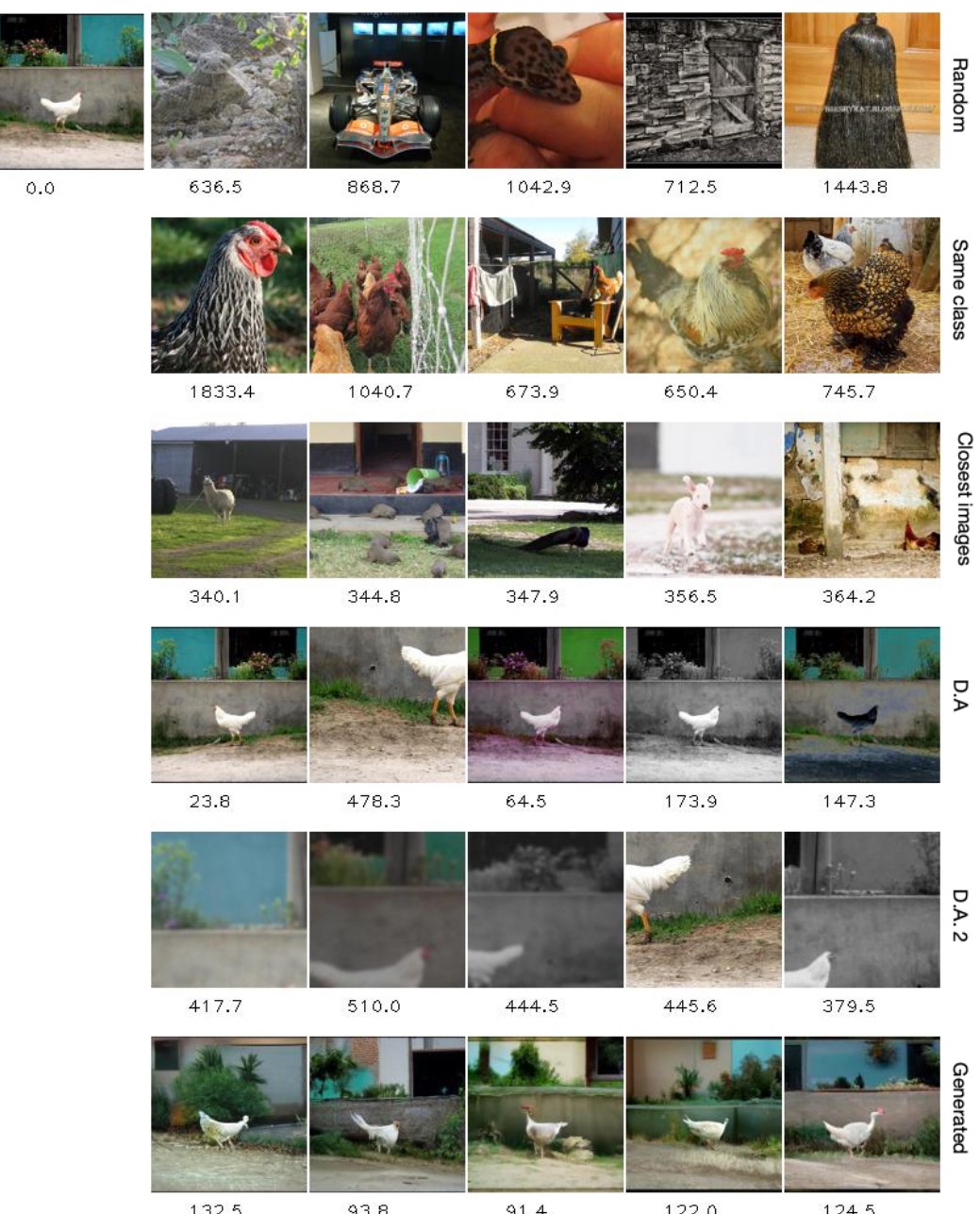

Figure 22: **Squared Euclidean distances in the Dino representation space.** We show the squared euclidean distance between the conditioning image on the leftmost column on first row and different images to get an insight about how close the samples generated by the diffusion model stay close to the representation used as conditioning. The distances with the conditioning is printed below each images. On the first row, we show random images from the ImageNet validation data. On the second row, we take random validation examples belonging to the same class as the conditioning. On the third row, we find the closest training neighbors of the conditioning in the representation space. On forth row, D.A. means Data Augmentation which consist in horizontal flip, CenterCrop, ColorJitter, GrayScale and solarization. On fifth row (D.A. 2), we use the random data Augmentation used in the paper of (Caron et al., 2020; 2021). On the last row, we show the generated samples from our conditional diffusion model that use **Dino representation**. The samples produces by our model are much closer to the conditioning than other images.

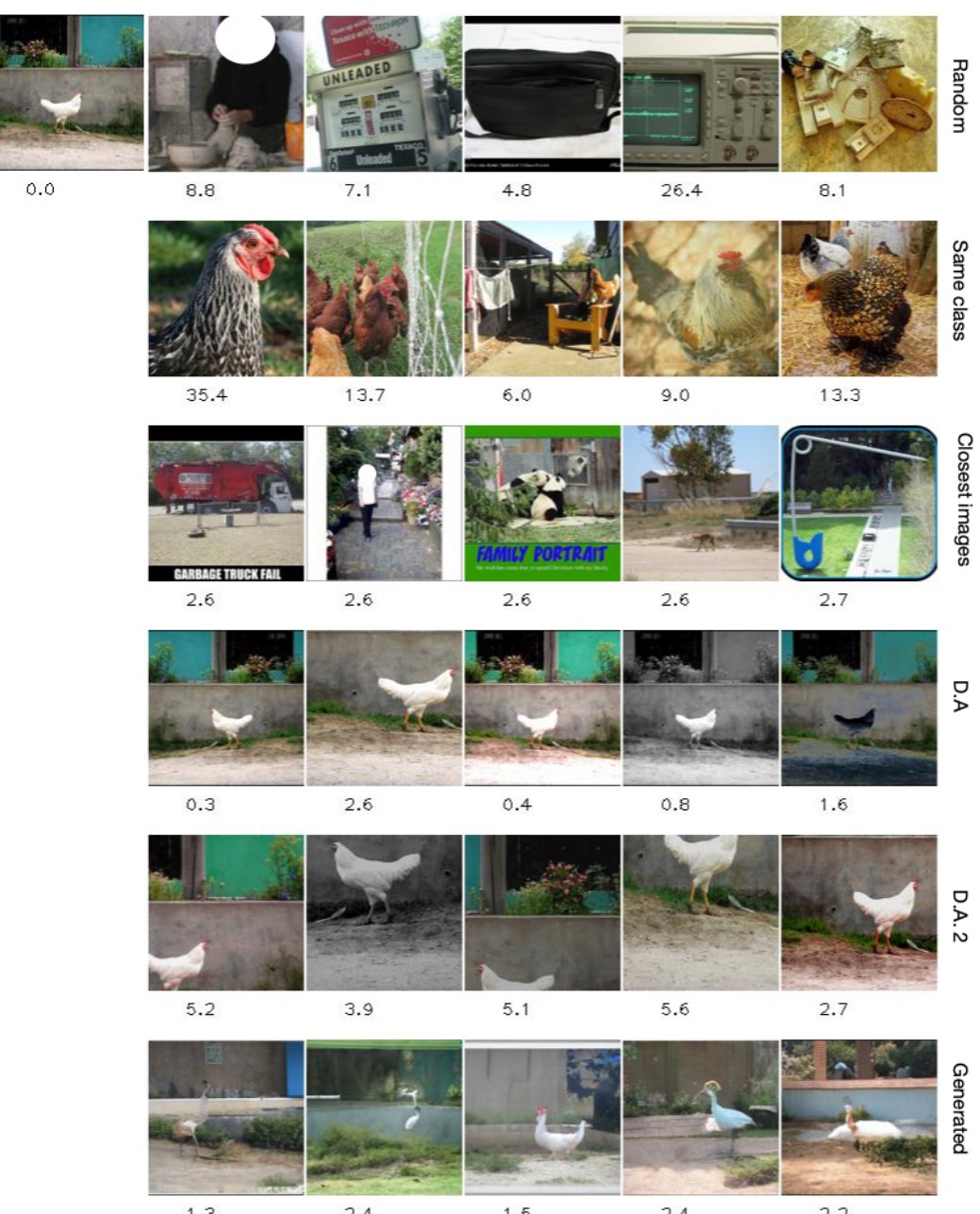

Figure 23: **Squared Euclidean distances in the SimCLR projector head representation space.** We show the squared euclidean distance between the conditioning image on the leftmost column on first row and different images to get an insight about how close the samples generated by the diffusion model stay close to the representation used as conditioning. The distances with the conditioning is printed below each images. On the first row, we show random images from the ImageNet validation data. On the second row, we take random validation example belonging to the same class as the conditioning. On third row, we find the closest training neigbords of the conditioning in the representation space. On forth row, D.A. means Data Augmentation which consist in horizontal flip, CenterCrop, ColorJitter, GrayScale and solarization. On fifth row (D.A. 2), we use the random data Augmentation used in the paper of (Caron et al., 2020; 2021). On the last row, we show the generated samples from our conditional diffusion model that use **SimCLR projector head representation**. The samples produces by our model are much closer to the conditioning than other images.

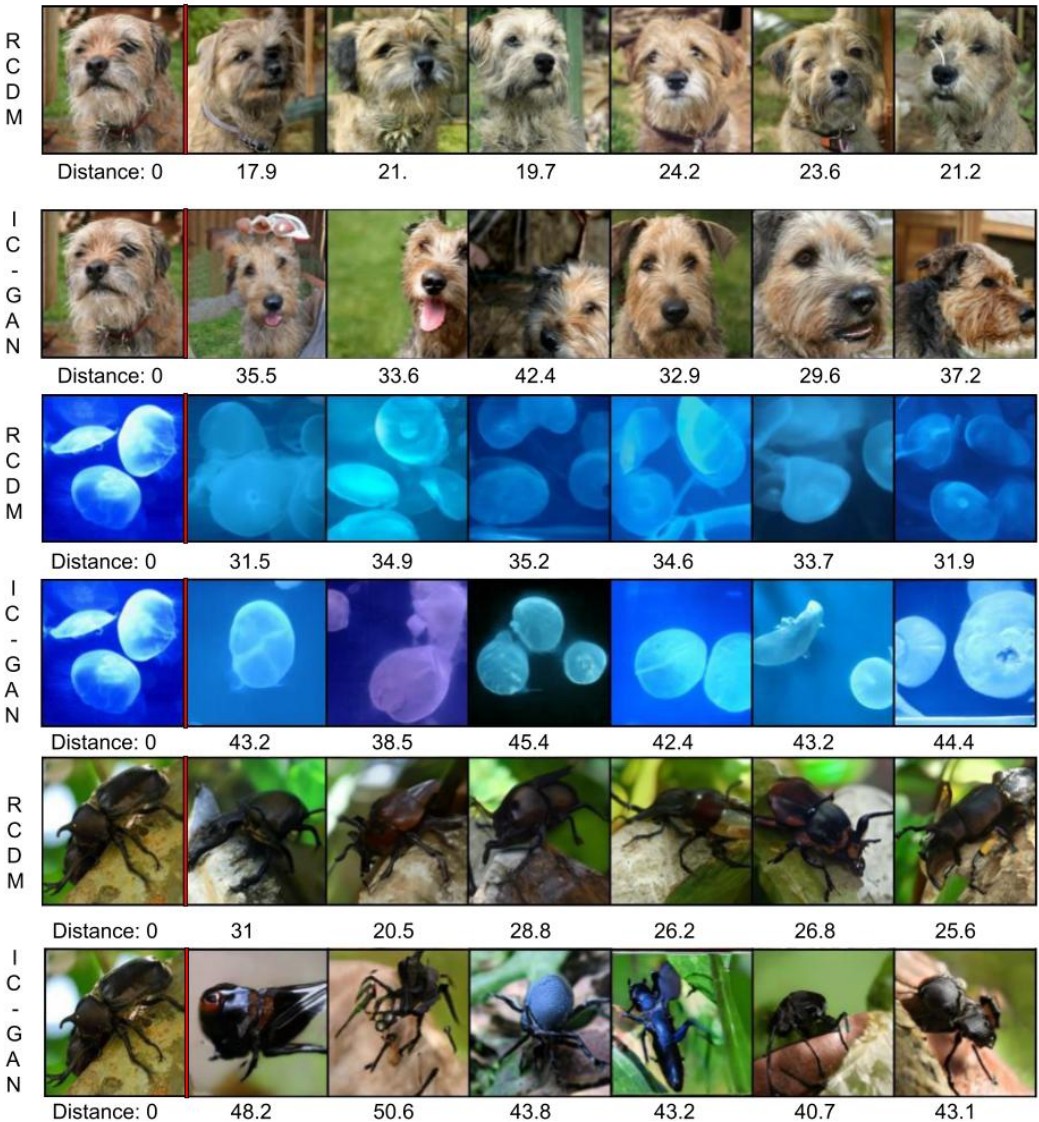

Figure 24: Comparison of the euclidean distance between IC-GAN and RCDM. We use the same self-supervised representation as conditioning (Swav encoder) for RCDM and IC-GAN. We compute the euclidean distance between the representation of the generated images versus the representation used as conditioning. We observe that samples of RCDM are much closer in the representation space (and also visually) to the conditioning. Samples of IC-GAN show a higher variability, thus farther away in the representation space.

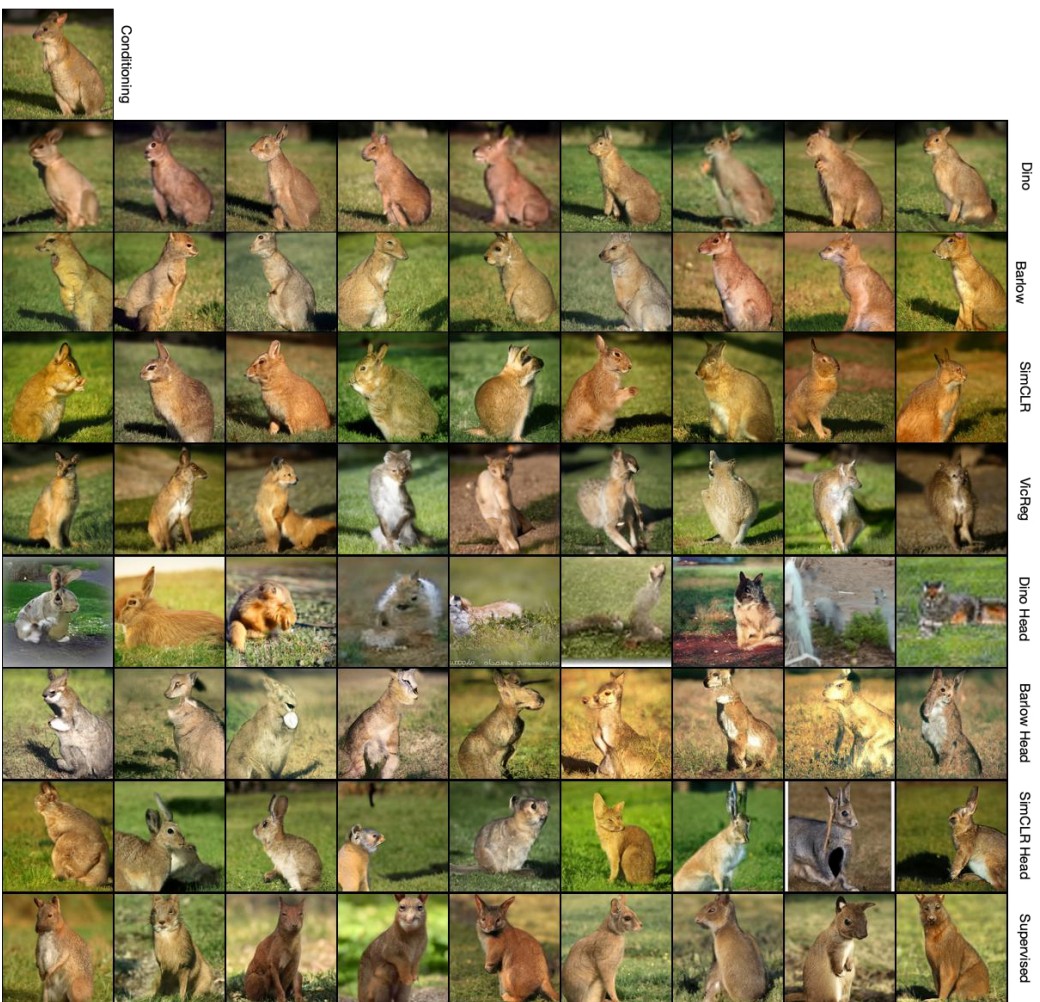

Figure 25: Generated samples from RCDM trained with representation from various self-supervised models. The image used for conditioning is a baby kangaroo on the top row, left-most column. Then, we generate 9 samples for each model with different random seed. We observe that the representation given by dino isn't very invariant while the one given by SimCLR or VicReg show much better invariance. We also show the samples of RCDM trained on the representation given by the projector (The embedding on which is usually applied the SSL criterion). There is a much higher variability in the generated samples. Maybe too much to be used for a classification task since we can observe class crossing.

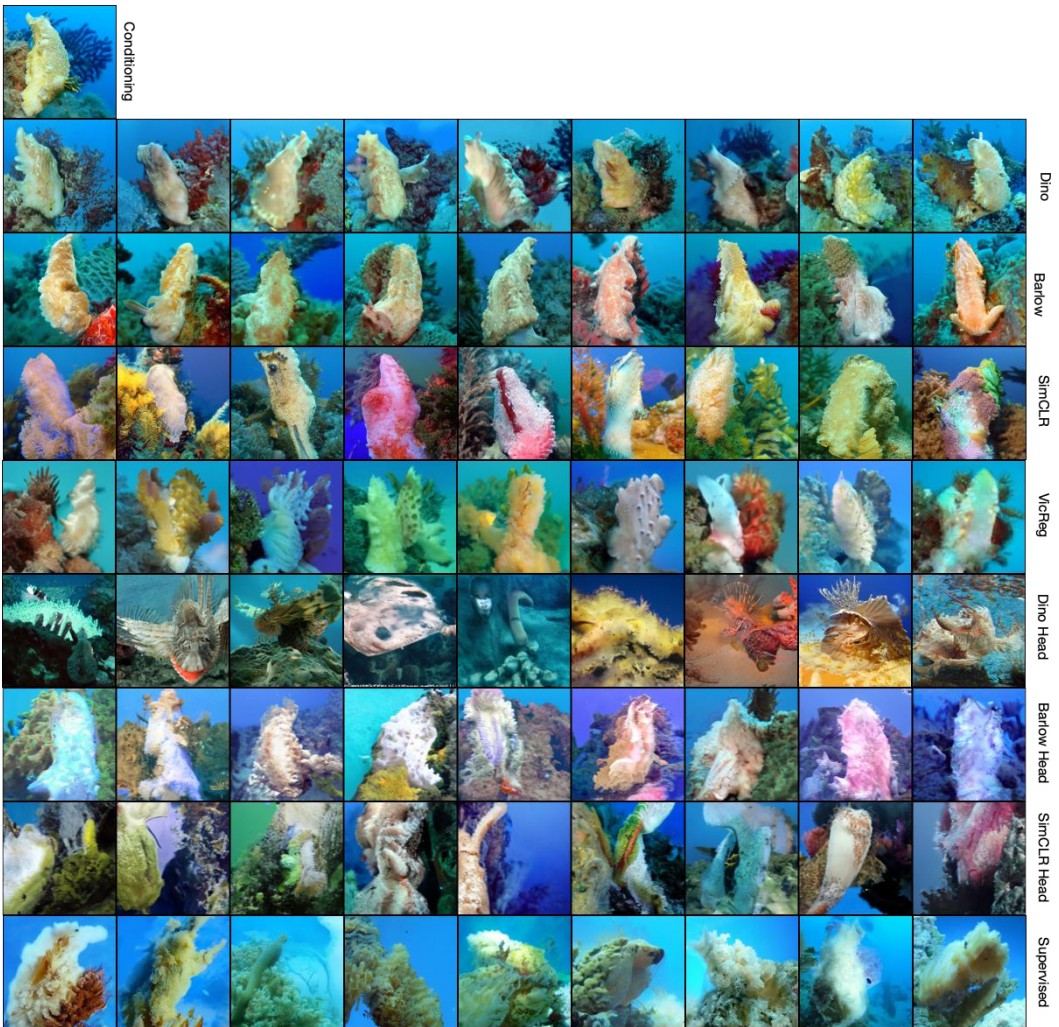

Figure 26: Generated samples from RCDM trained with representation from various self-supervised models. We generate 9 samples for each model with different random seeds. We observe that the representation given by dino isn't very invariant while the one given by SimCLR or VicReg show much better invariance. We also show the samples of RCDM trained on the representation given by the projector (The embedding on which is usually applied the SSL criterion). There is a much higher variability in the generated samples. Maybe too much to be used for a classification task since we can observe class crossing.

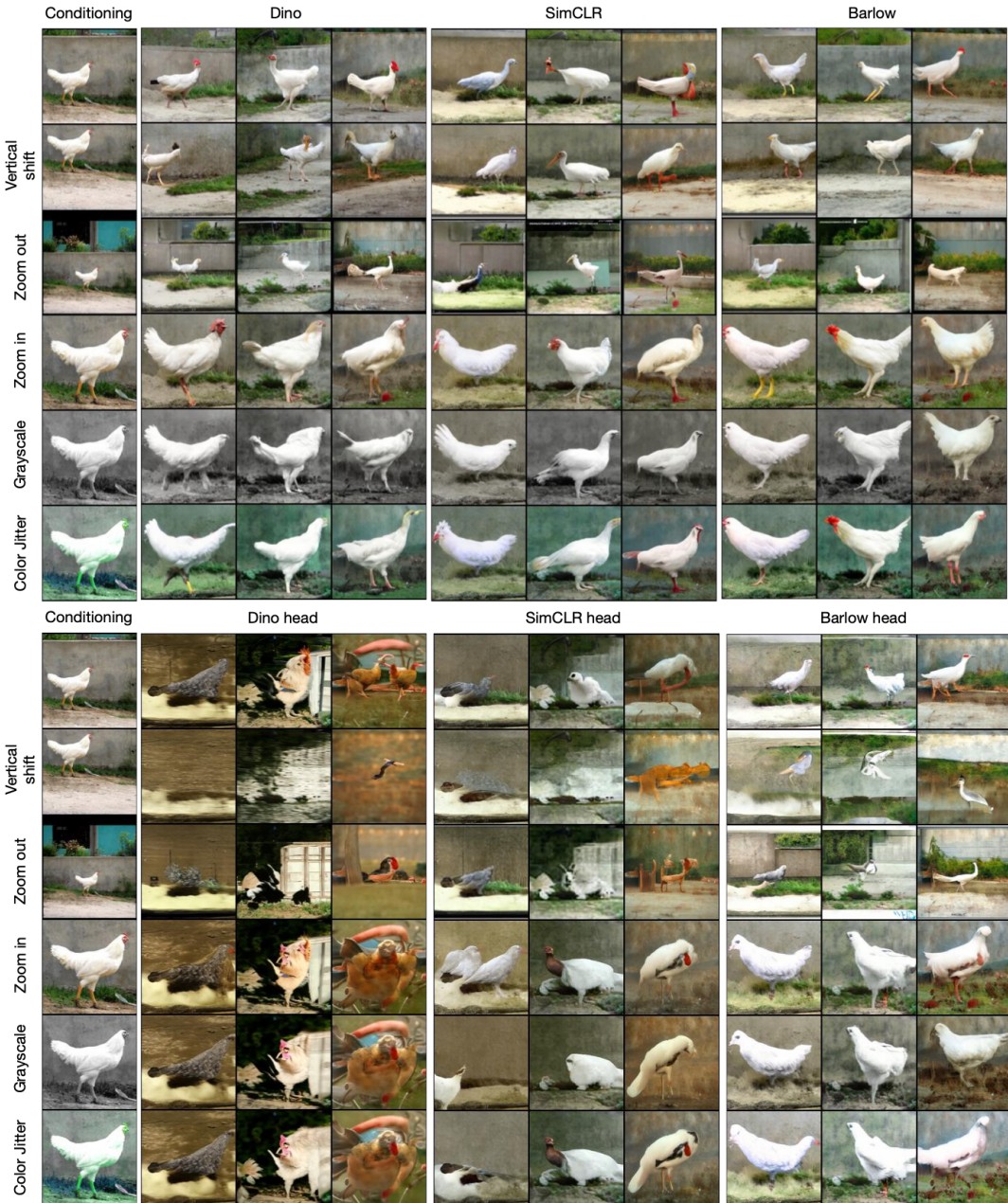

Figure 27: We compare how much the samples generated by RCDM change depending on different transformations of a given image and the model and layer used to produces the representation. Top half uses 2048 representation. Bottom half uses the lower dimensional projector head embedding. We observe that using the projector head representation leads to a much larger variance in the generated samples whereas using the traditional backbone (2048) representation leads to samples that are very close to the original image. We also observe that the projector representation seems to encode object scale, but contrary to the 2048 representation, it seems to have gotten rid of grayscale-status and background color information.

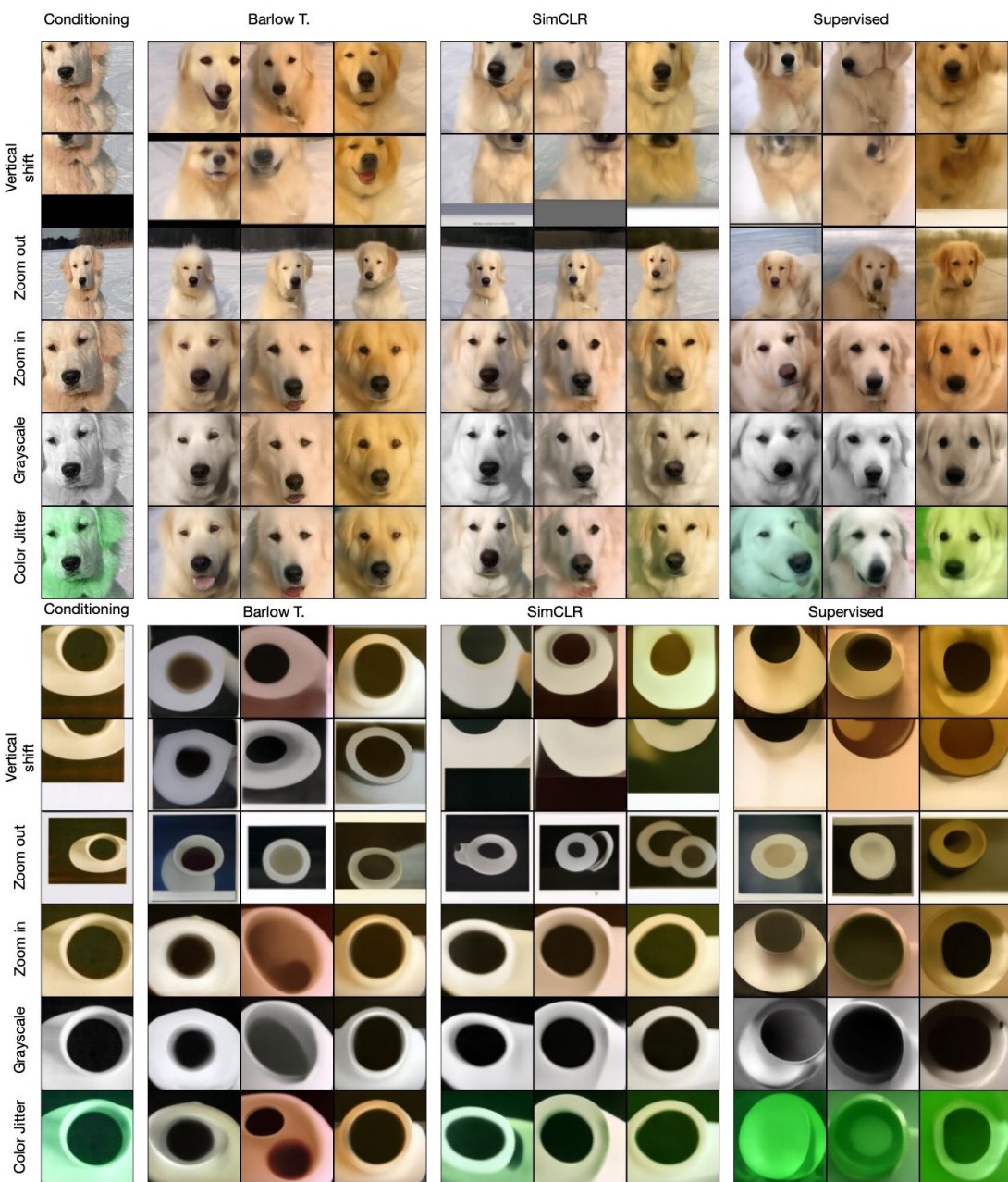

Figure 28: Same setup as Figure 27 except with other images as conditioning.

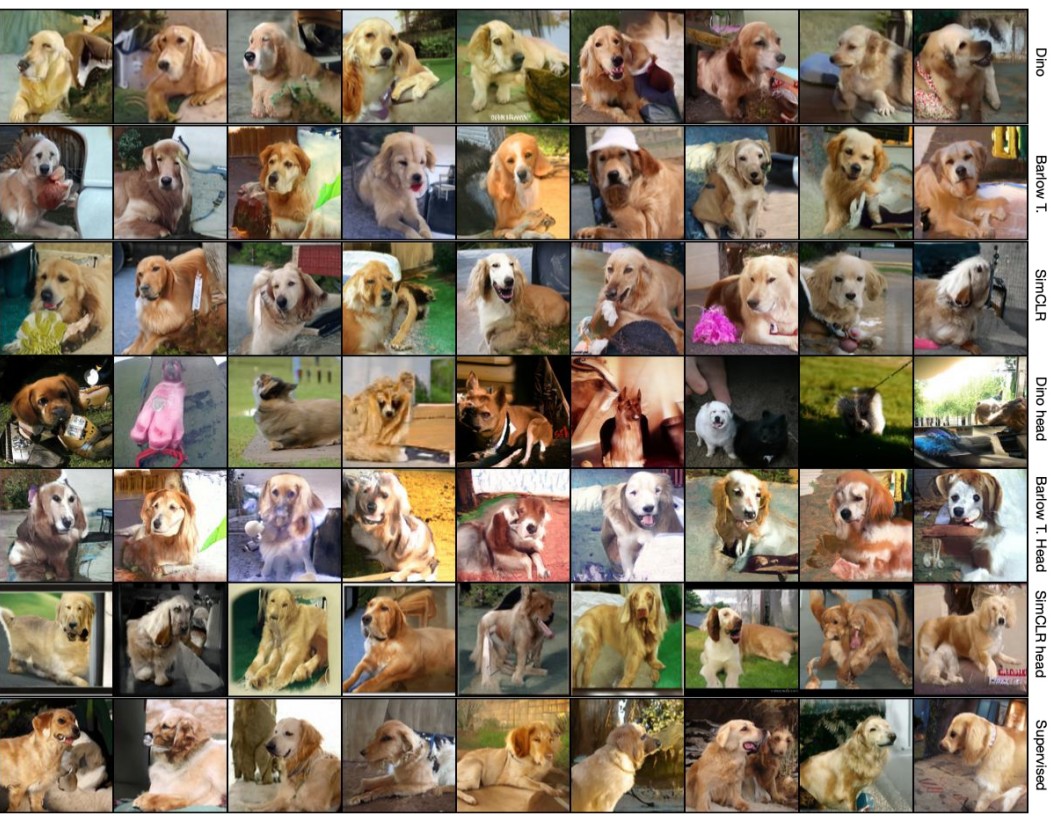

Figure 29: Generated samples from RCDM using the mean representation for a specific class (golden retriever) in ImageNet for various SSL models.

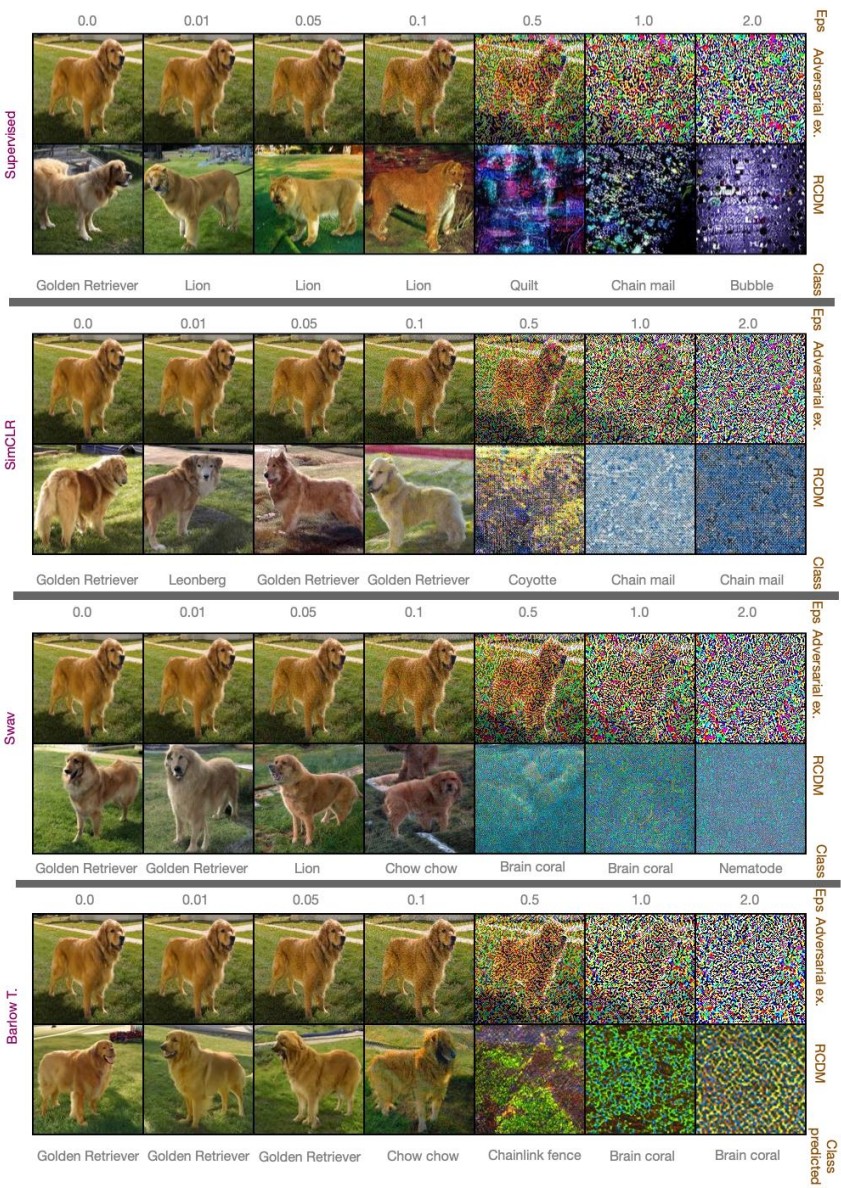

Figure 30: **Visualization of adversarial examples** We use RCDM to visualize adversarial examples for different models. For each model, we trained a linear classifier on top of their representations to predict class labels for the ImageNet dataset. Then, we use FGSM attack over the trained model using a NLL loss to generate adversarial examples towards the class lion. For each model, we visualize adversarial examples for different values of $\epsilon$ which is the coefficient used in front of the gradient sign. In the supervised scenario, even for small values of epsilon which doesn't seem to change the original image, the decoded image as well as the predicted label by the linear classifier becomes a lion. However it's not the case in the self-supervised setting where the dog still get the same class or get another breed of dog as label until the adversarial attack becomes more visible to the human eye (For $\epsilon$ value superior to 0.5).

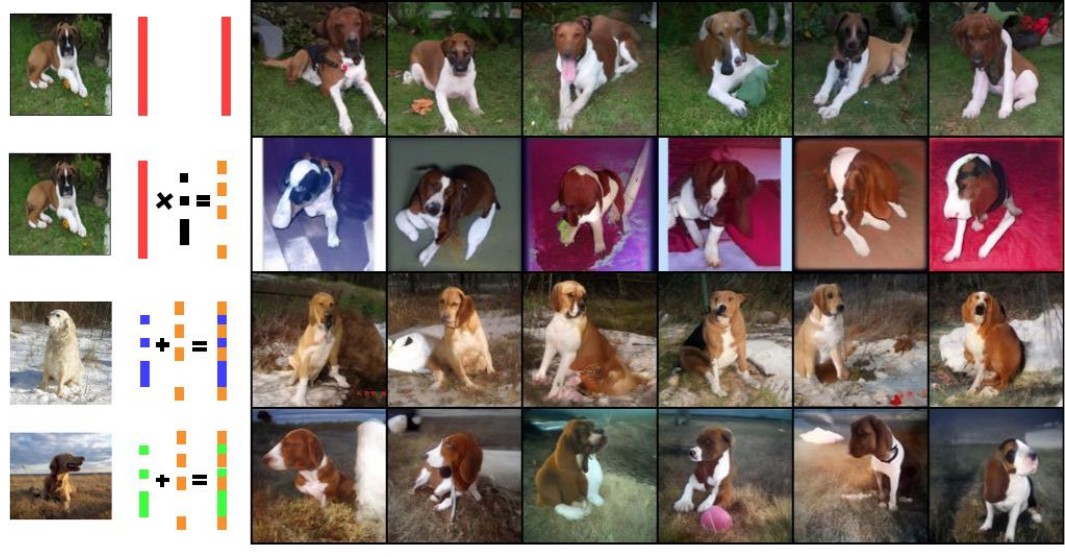

Figure 31: **Background suppression and addition** Visualization of direct manipulations over the representation space. On the first row, we used the full representation of the dog's image on the top-left as conditioning for RCDM. Then, we find the most common non zero dimension across the neighborhood of the image used as conditioning. On the second row, we set these dimensions to zero and use RCDM to decode the truncated representation. We observe that RCDM produces examples of the dog with a high variety of unnatural background meaning that all information about the background is removed. In the third and forth row, instead of setting the most common non zero dimension to zero, we set them to the value of corresponding dimension of the representation associated to the image on the left. As we can see, the original dog get a new background and a new pose.

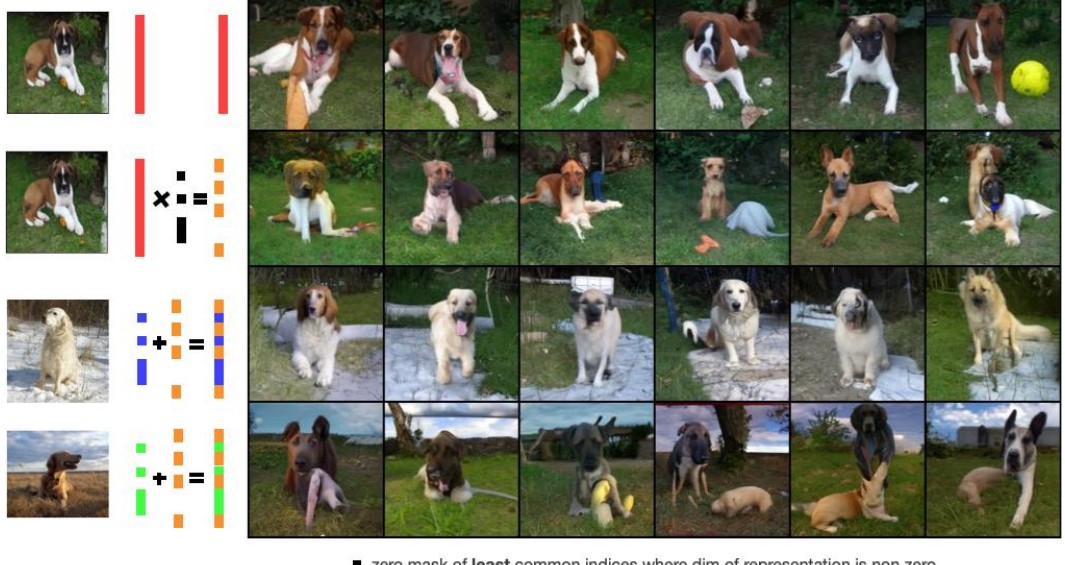

Figure 32: Same setup as Figure 31 except that instead of using the most common non zero-dimensions as mask, we used the least common non-zero dimensions as mask. On the second row, we observe that some information about the original dog is removed such that in each column, we get a slightly different breed of dog while the background stay fixed. On the third and forth row, we saw that the information about the background (grass) is propagated through the samples (which was not the case in Figure 31).

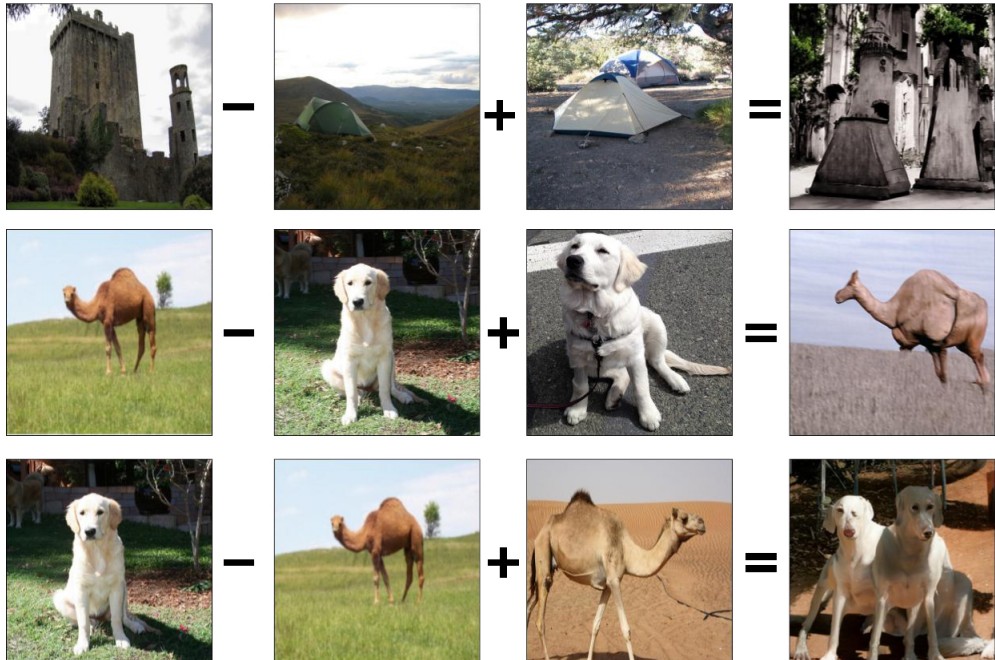

Figure 33: Algebraic manipulation of representations from real images (left-hand side of =) allows RCDM to generate new images with novel combination of factors. Here we use this technique with ImageNet images, to attempt background substitutions.

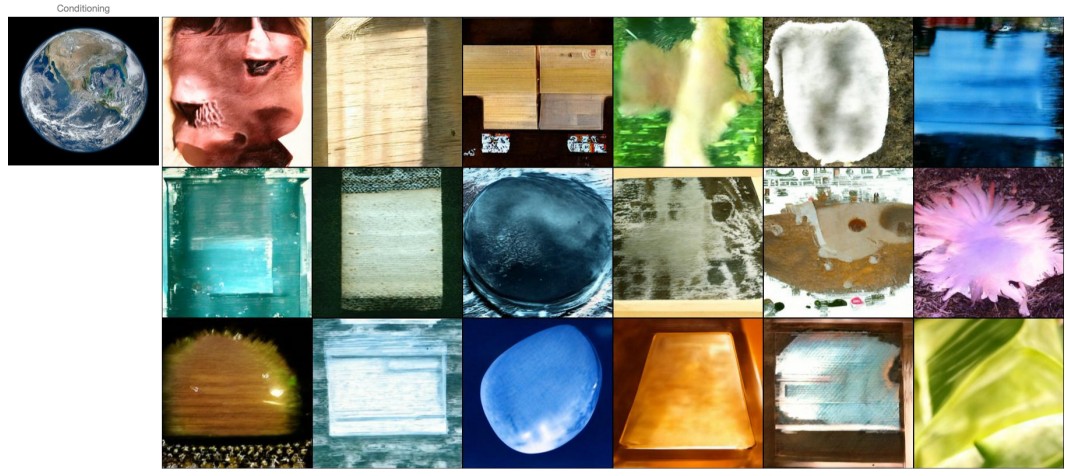

(a) Earth from an untrained representation (Random initialized Resnet 50).

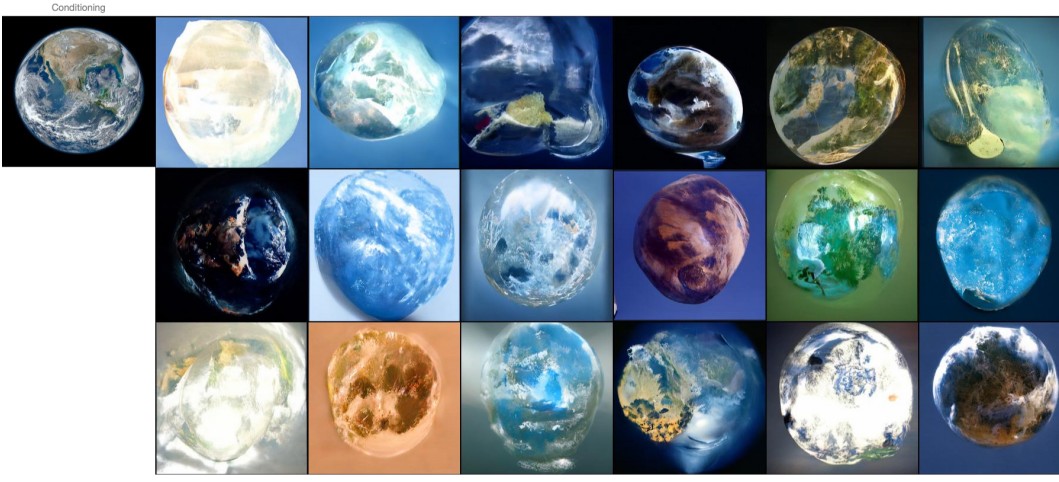

(b) Earth from a supervised representation (Pretrained resnet50 on ImageNet)

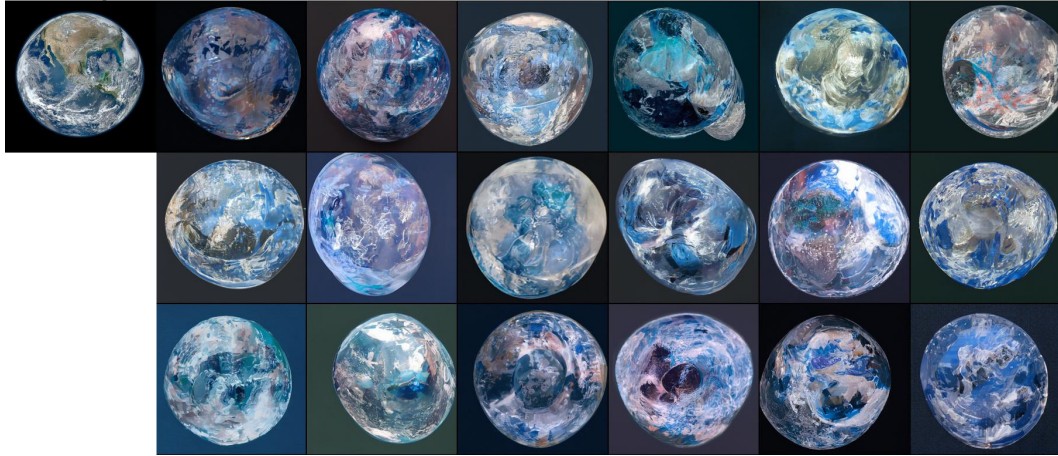

(c) Earth from a SSL representation (Dino Resnet50 backbone).

Figure 34: Different samples of RCDM conditioned on a satellite image of the earth (source: NASA). We show the samples we obtained in a) when using a random initialized network to get representations, b) when using a pretrained resnet50, c) when using a self supervised model (Dino).

