# OpenReview forum: "High Fidelity Visualization of What Your Self-Supervised Representation Knows About"
_ICLR.cc/2022/Conference — ICLR 2022 Submitted_

### Official Review · Reviewer_mFTx · 2021-10-31

**Correctness:** 4
**Technical Novelty And Significance:** 3
**Empirical Novelty And Significance:** 4
**Recommendation:** 8
**Confidence:** 3

**Main Review:**

**Strength**: This work provides a visualization tool for understanding NN representations. The proposed RCDM is not technically challenging (it is adapted from ADM in the prior work), but it works well for probing the information in NN representations without sacrificing generation quality.

The experiments cover different SSL methods and the comparison with supervised learning, showing results on 1) in/out of distribution generation, 2) interpolation, 3) super-resolution, 4) unconditioned generation, 5) algebraic manipulation.

Other aspects:
  - The writing is clear and easy to follow.
  - The relation and comparison with prior work is discussed adequately.
  - The code will be released.

**Questions**:
  - Figure 1: it would be better to provide more context for the relative distance: e.g. is the difference between 0.4% and 3.3% significant?  Should a 3% distance considered small?
  - Figure 6: projector head vs backbone: are features from the projector able to get as close to the original feature as features from the backbone? If not, how much further are the projector features compared to the backbone features?

Minor comments:
  - The writing for Figure 7 is repetitive, i.e. the last paragraph on page 8 and the caption for Figure 7.
  - last sentence in the first paragraph: "...or discriminating instances."
  - the paragraph below equation (2): should the dimension be $K$ rather than $S$?
  - Sec 4.1, second line in the second paragraph: model "dependent"
  - Last sentence of the caption of Figure 4: should be column 2 to 6 (rather than 7)
  - Appendix section C: it should link to Figure 15 rather than Figure 21.



**Summary Of The Paper:**

This paper proposes a conditional diffusion model that can be used to visualize representations learned by SSL or supervised models.

The proposed Representation-conditioned Diffusion Model (RCDM) can generate images that are both close in the representation space to a given image, and looking realistic.

The paper also provides empirical results comparing conditionally generated samples from models trained on features from the backbone or projector of a SSL model, as well as supervised features. These experiments provide insight to understanding SSL, such as showing that projector is the key of SSL being invariance to transformations.


**Summary Of The Review:**

The method in this paper is unsurprising but gives good results.
I find the experiments thorough and insightful, and would recommend an accept.

---

> ### Author Response · Authors · 2021-11-23
> **Answer to reviewer mFTx**
>
> Thank you for your time and your careful review. Please first read our “common answer” above. We now answer your specific comments:
> We corrected the typos you mentioned. Thank you very much !
>
>
> *"Figure 1: it would be better to provide more context for the relative distance: e.g. is the difference between 0.4% and 3.3% significant? Should a 3% distance considered small?''*
>
> The main issue is that it is difficult to have a ground truth or reference in terms of distances. It mostly depends on the scale which can vary a lot between models. Note that we moved this subsection (including figure 1) on gradient based optimization for conditional generation to the appendix, as most results were explored in previous studies. We instead focused on our core contributions: obtaining insights into deep networks’ representation contents.
>
> *"Figure 6: projector head vs backbone: are features from the projector able to get as close to the original feature as features from the backbone? If not, how much further are the projector features compared to the backbone features?''*
>
> It is hardly possible to meaningfully compare distances between the backbone and the projector directly as these are different spaces of largely differing dimensions using different normalizations and scales.

---

> > ### Comment · Reviewer_mFTx · 2021-11-29
> > **Thank you for the clarifications**
> >
> > I thank the authors for their clarifications.
> >
> > Regarding changes mentioned in the general response, I'd suggest adding a reference in the main text to Fig 11 (previously Fig 3), since I think it's a nice figure that can help clarify the models.
> > Also, a super minor point: in your general response to authors, under the second bullet point for "List of changes", should "Fig 1" be "Fig 3"?

---

### Official Review · Reviewer_YLR1 · 2021-11-01

**Correctness:** 4
**Technical Novelty And Significance:** 2
**Empirical Novelty And Significance:** 3
**Recommendation:** 5
**Confidence:** 4

**Main Review:**


This paper considers a very worthy problem, that of analyzing encoders via visualizations, and clearly develops a high quality generative modeling approach with which to do this. I think this paper may be on its way to being a vey nice paper, however I have concerns with it’s current state.

Current strengths include:

- high quality of generative models used (the generated samples are high fidelity).
- plenty of qualitative evaluation - the appendix contains many pages of additional visualizations.
- the method is able to visually discern some differences between supervised and self-supervised trained encoders. The differences between  self-supervised methods seems a little harder to extract, besides noting that some (e.g., DINO) exhibit better invariance than others (e.g., SimCLR).


My two broad concerns (i.e., "weaknesses"), which I discuss in more detail momentarily, are:

- A lack of *quantitative results*. All results presented are qualitative and visual in nature (with the exception of a sanity check that the generative models are producing photorealistic samples).
- A lack of demonstration of useful applications for the proposed method.

**Quantitative results:**

While a large part of generative modeling is focused on qualitative production of photorealistic samples, in order to properly compare you method and analyze pertained encoders I think it is critical to have more quantitative evaluation.

I am sure the authors can come up with many better ideas for additional quantitative evaluations than I can, but I will also attempt to offer some actionable feedback:

- The main premise of the work is that the samples $x’$ from your diffusion model are such that $f(x’)$ are very close to the original representation h. It seems almost a must that this is quantitatively confirmed.
- You argue verbally that for a given representation $h$, IC-GAN generates samples which map to a neighborhood of $h$, whereas your method generates samples that map very closely to $h$. While this may very well be true, it would be much more convincing if you quantitatively showed this.

**Applications of method:**

At present, the work mainly attempts to visualize some simple properties that one would expect a self-supervised encoder to have - e.g., a more instance-level invariance then supervised encoders, and invariance to the data augmentations used training training. Unfortunately I don't find this to be sufficiently insightful right now. In order to truly demonstrate the value of this method, it would be very valuable to take more steps towards devising practically useful tools that are enabled by this method. Some ideas of the kind of thing I am imagining are:

- Could your visualization method be used to determine a priori which out of a group of encoders will perform best at a given downstream task. If possible, this would be very cool since the trial and error evaluation of pretrained encoders is annoyingly heuristic. E.g. specifically, could you develop some metric based on your visualization method that correlates well with (e.g.,) segmentation performance.
- Suppose you have two encoders $f$ and $f’$, and you know that f performs much better on some downstream task $T$. Could you use the diffusion model for $f$ to *generate new training samples* with which to finetune the encoder f’ itself (not just the linear probe) so that the performance of $f’$ on task $T$ improves.


To be clear, I am not asking you to try any of these ideas specifically. Instead I am trying to illustrate the type of thing I envisage when I ask for extensions that analyze encoders in more detail. This is a promise that was made in the abstract — “new avenues to analyze and improve self-supervised models” — but which was underdelivered on in my eyes.

---
**Miscellaneous comments and questions:**

- The difference in FID and IS for your method compared to ADM seems very large considering your method is a modification of theirs. Do you know what accounts for this? Is it better model training? Is it the fact that your method is instance conditioned, & therefore requires less variation in samples? Something else?
- Some experiments seem orthogonal to the idea of analyzing pretrained encoders. E.g. the value of Fig 4c) is unclear to me (4 a) and b) are good) and seem to be more of the form of a “sanity check” that the generative model is working sensibly.
- Please fix your bibliography. Many citations are missing their publication venues (e.g. Barlow Twins and at least three of Yang Song’s papers), and some citation simply list: 1) authors names, 2) paper’s name, 3) year — i.e. they don’t even have an arXiv number, let alone a publication venue (e.g. SimCLR and VICReg papers).



**Summary Of The Paper:**

This work introduces develops a method for sampling different natural images that a pretrained encoder maps to the same or similar representation. To do this, the authors propose a modification to the diffusion model of Dhariwal & Nichol for sampling conditioned on a given encoded representation $h=f(x)$ that is well suited to conditioning on high dimensional vectors (i.e., representations). This new method is used to qualitatively compare encoders trained via a number of methods (supervised, DINO, SimCLR, Barlow Twins, VICReg etc.).



**Summary Of The Review:**


The reasons I am concerned about these two main objections are:
- quantitative results are almost a must in order to properly evaluate methodological progress, and  check that the method is behaving as it should, and to properly compare models/methods etc.,
- The methodological contribution of the representation conditioned diffusion model is relatively small (it sufficed to simply describe the changes one needs to make to ADM in words). This is not a problem in itself. But I do think it is therefore important to demonstrate the method's usefulness in analyzing and understanding pretrained models.

Consequently, I am currently not in favor of acceptance. However, to recognize the generally high quality of the work thus far I am opting for a weak reject. I am unlikely to raise the score to an accept without significant updates to the work, or strong arguments in favor from other reviewers.

---

> ### Author Response · Authors · 2021-11-23
> **Answer to reviewer YLR1 (2/2)**
>
> *“ Instead I am trying to illustrate the type of thing I envisage when I ask for extensions that analyze encoders in more detail. This is a promise that was made in the abstract — “new avenues to analyze and improve self-supervised models” — but which was underdelivered on in my eyes.”*
>
> When we present RCDM as a way to analyse and improve self-supervised models, it is with respect to the current constraint of a self-supervised setting. Many researchers in SSL have to use artificial and simple datasets to measure how successful their SSL criteria are at making the representation invariant to specific and complex factors of variations.  However, there is a gap between the experiments and results we can get on dataset like colored-MNIST with labelled factor of variations and natural images with dataset like ImageNet which can’t have every pixel labelled for each possible factor of variation. With a stochastic decoder like RCDM, researchers will be able to visually understand how much their SSL criteria is working for any factor of variation without the need for an extensive and costly labelling process. As an example, with our method, it is very straightforward to see how much background information is contained or not in the representation. So we really believe that the method we suggest brings new avenue because it gives a way for researchers to gain more insight about the representation learned than just having a class accuracy score (which doesn’t reflect the invariance to background or spurious correlation) or other metric that would requires expensive labelling and mathematical approximations.
>
> *“ The difference in FID and IS for your method compared to ADM seems very large considering your method is a modification of theirs. Do you know what accounts for this? Is it better model training? Is it the fact that your method is instance conditioned, & therefore requires less variation in samples? Something else?”*
>
> We think that instance conditioning is helping the sampling chain to converge towards specific examples. It’s easier to modelize and learn p(x | h) than purely p(x). It's similar to the method based on classifier guidance which learn p(x | y) (where y is a class label) which biases the sampling chain towards more probable examples under a specific class by following a classifier gradient.
>
> *“Some experiments seem orthogonal to the idea of analyzing pretrained encoders. E.g. the value of Fig 4c) is unclear to me (4 a) and b) are good) and seem to be more of the form of a “sanity check” that the generative model is working sensibly.”*
>
> You are right, all of these experiments were sanity checks. The value of the interpolations figure you are mentioning is to show that our model doesn’t just overfit specific values of h but can generalize to a new and never seen combination of two differentes representations. This also visually proves that the generated images are faithful to the representation (if not the generated image will not share any characteristics with the images used for the interpolation).
>
> *“Please fix your bibliography. Many citations are missing their publication venues (e.g. Barlow Twins and at least three of Yang Song’s papers), and some citation simply list: 1) authors names, 2) paper’s name, 3) year — i.e. they don’t even have an arXiv number, let alone a publication venue (e.g. SimCLR and VICReg papers).”*
>
> Thank you, we fixed that !

---

> ### Author Response · Authors · 2021-11-23
> **Answer to reviewer YLR1 (1/2)**
>
>
>
>
> Thank you for your time and your careful review. Please first read our “common answer” above. We now answer your specific comments:
>
>
> *“lack of quantitative results”*
>
> The aim of our paper is precisely to propose, showcase, and highlight the merits of a *qualitative* visual analysis method. So quantitative results are understandably of limited relevance here: they are not the point.
>
> Following your suggestion, we however added new relevant quantitative results in the main paper, to measure how close the representation of generated samples are from the original conditioning representation h (Figure 2, b).
>
> *“demonstration of useful applications for the proposed method.”*
>
> We added two new applications as examples in the main paper: visualization of adversarial examples (Sect 5.2, Fig. 5 and Fig. 30), and algebraic manipulation of representations (Sect 5.3 Fig. 6, 31, 32) . The last application provides novel insights about how the representation is structured locally and which factors of variations are the least and most common across a neighborhood of representations.
>
> *“The main premise of the work is that the samples x′ from your diffusion model are such that f(x′) are very close to the original representation h. It seems almost a must that this is quantitatively confirmed”*
>
> We quantitatively confirmed this in Figure 2, b).
>
> *“You argue verbally that for a given representation h, IC-GAN generates samples which map to a neighborhood of h, whereas your method generates samples that map very closely to h. While this may very well be true, it would be much more convincing if you quantitatively showed this.”*
>
> We quantitatively confirmed this in Figure 24.
>
> *“Could your visualization method be used to determine a priori which out of a group of encoders will perform best at a given downstream task. If possible, this would be very cool since the trial and error evaluation of pretrained encoders is annoyingly heuristic. E.g. specifically, could you develop some metric based on your visualization method that correlates well with (e.g.,) segmentation performance.”*
>
> Yes, it is possible ! For example, with RCDM one can visualize if the representation contains information about the pose of an object. If this information is thrown away (meaning that RCDM generates samples with different poses), the representation will not be very useful for any task related to pose estimation. We could also have the case where a specific representation contains information about the pose of a dog without containing information about its breed. In this case, a classification task will fail whereas the pose estimation will probably work well. The issue with developing a specific metric is that it will probably require using labeled data. To quantify how much the pose of an object is changing through different samples, one would need to have data with pose labels for the object.
>
> *“Suppose you have two encoders f and f′, and you know that f performs much better on some downstream task T
> . Could you use the diffusion model for f to generate new training samples with which to finetune the encoder f’ itself (not just the linear probe) so that the performance of f′ on task T improves”*
>
> This is an interesting idea to try in future work! If the performance of f is better at a given task, it implies that the learned representations of f are less sensitive to the nuisance variables present in the dataset (and irrelevant to the task at hand) than the representations of f’. Additionally, f maintains a greater amount of information about the task at hand than f’. This means that RCDM conditioned on f, will produce samples that vary w.r.t. those nuisance variables (as they are the invariant space of f). Hence, doing data-augmentation with those samples, would force f’ to also adapt its invariant space accordingly. In the limit, f and f’ should reach identical performances, assuming that the approximation power, and inductive biases of f and f’ are similar.

---

### Official Review · Reviewer_rQaS · 2021-11-02

**Correctness:** 3
**Technical Novelty And Significance:** 2
**Empirical Novelty And Significance:** 2
**Recommendation:** 6
**Confidence:** 4

**Main Review:**

The main idea of the paper, i.e. explaining the representations (and their invariances) learned by deep neural networks via visualizations through natural images is intuitive and very reasonable. By demonstrating that gradient-based methods do not suffice (Sec. 2) to reconstruct _natural_ images, the authors show the need for a probabilistic approach and, by opting for a conditional diffusion model, provide a method capable of generating high-quality images corresponding to the representations, as demonstrated in the experimental section. The main drawback of this paper, however, is that this idea is not novel and the paper fails to discuss prior work which uses different implementations for the conditional generative model, e.g. GANs [1], autoregressive models [2] or normalizing flows [3]. Authors should clearly situate their work within this existing literature and discuss what advantages/differences an approach based on DDPMs offers. For example, is it possible to use a deterministic approximation of the diffusion process (such as DDIM, see [4]) to extract, analogously to [3], an explicit representation of the learned invariances? Further, I do not quite understand the focus on SSL representations, as the proposed techniques can easily be applied to any type of representation, independent of the training paradigm.

Moreover, a quantitative assessment of the correspondences between representations and synthesized images is missing. For example, can the synthesized representations be fed back into the (SSL) model and does this then yield the same representation $h$?

_Mixed comments:_

- I think the paper would benefit from focusing on __explaining__ (SSL) representations and thus dropping the part on synthesis (mainly Appendix B), as it mainly relies on methods that already have been developed and discussed in the context of pure synthesis (ADM).
- On the contrary, Appendix E and especially Fig. 20 should be moved to the main paper, because they provide interesting analysis of different SSL models/representations.
- 1st paragraph of 4.2.: I do not quite understand the motivation here: In the case the density of $p(x|h)$ approaches a delta-distribution, the task becomes more easy and in the limit, could be implemented through a regression model.
- Following on from this: the paper should demonstrate the need for a probabilistic approach by comparing, for example, with a simple regression baseline $x = f_{\theta}(h)$. A more advanced baseline would be the IC-GAN discussed in Sec. 4.2 (without neighborhood sampling).
- _"Diffusion models do not usually explicitly express an energy function"_ (Footnote 4) The usual training mechanism is to train the reweighted epsilon-parameterization from [5] which corresponds to a denoised score matching objective. This score corresponds to the derivative of an energy which can be plugged into MCMC sampling, see [6].
- The figure on the first page should show several examples of each model to emphasize the visualization capability of the invariances.
- The interpolation in Fig. 4c should show several "interpolation-paths", demonstrating the stochasticity of the proposed approach (DDPM)
- Does the proposed conditioning mechanism via conditional batch norm provide significant advantages over other methods (e.g. addition to the timestep-embedding, AdaIN, cross-attention, ...)?

__References:__
- [1]: Shocher, A., Gandelsman, Y., Mosseri, I., Yarom, M., Irani, M., Freeman, W.T., Dekel, T.: Semantic Pyramid for Image Generation.
- [2]: Nash, C., Kushman, N., Williams, C.K.: Inverting Supervised Representations with Autoregressive Neural Density Models.
- [3]: Rombach, R., Esser, P., Ommer, B.: Making Sense of CNNs: Interpreting Deep Representations & Their Invariances with INNs
- [4]: Song, J., Meng, C., Ermon, S.: Denoising Diffusion Implicit Models
- [5]: Jonathan Ho, Ajay Jain, and Pieter Abbeel. Denoising Diffusion Probabilistic Models
- [6]: Yang Song, Jascha Sohl-Dickstein, Diederik P. Kingma, Abhishek Kumar, Stefano Ermon, and Ben Poole. Score-Based Generative Modeling through Stochastic Differential Equations


**Summary Of The Paper:**

This paper proposes a technique for visualizing the representations of fixed pretrained self-supervised neural networks. More specifically, by using a conditional denoising diffusion probabilistic model (DDPM, coined _RCDM_ here), a learned neural network representation $h$ can be mapped (back) to image space (i.e., the space of "natural images") and, by drawing multiple samples conditioned on the same $h$, the invariances learned by the (self-supervised) network can be visualized. Experiments with different pre-trained models show that the method is indeed able to synthesize high quality natural images corresponding to the learned representations.

**Summary Of The Review:**

I think that the present paper deals with a very important problem, namely the "understanding" of learned neural representations. The proposed approach to do this by synthesizing "natural" images generated by a generative model conditioned on these learned representations (here a diffusion model) is intuitive and reasonable. The generated samples are of high quality and seem to be in agreement with the representations.

Unfortunately, this approach is not novel (only the use of a diffusion model instead of {GANS, ARMs, flows} is new here) and the present work would, in my opinion, benefit greatly from being placed in the context of this existing work. In addition, there are some specific questions (see above) that I think are not yet answered satisfactorily.

++++++++++++++++++++++++++

Score raised to 6 after the rebuttal

---

> ### Author Response · Authors · 2021-11-23
> **Answer to reviewer rQaS (3/3)**
>
> *“I think the paper would benefit from focusing on explaining (SSL) representations and thus dropping the part on synthesis (mainly Appendix B), as it mainly relies on methods that already have been developed and discussed in the context of pure synthesis (ADM).”*
>
> You are entirely right, the specific synthesis method we chose to use is not a primary contribution, we updated the paper to emphasize the SSL representation part.
>
> *“1st paragraph of 4.2.: I do not quite understand the motivation here: In the case the density of  p(x|h) approaches a delta-distribution, the task becomes more easy and in the limit, could be implemented through a regression model. Following on from this: the paper should demonstrate the need for a probabilistic approach by comparing, for example, with a simple regression baseline  x=fθ(h) . A more advanced baseline would be the IC-GAN discussed in Sec. 4.2 (without neighborhood sampling)."*
>
> In SSL methods particularly, the training criterion encourages invariance of the learned representation to transformations of the input image s.a. translations and scalings (through random-resize-crop augmentations). This is in addition to the strong dimensionality reduction from x to h. So p(x|h) is not a dirac delta: given h, there are many possible input images that could yield it. We had tried deterministic regression in early experiments, and it yielded outputs so strongly blurred they were no longer recognizable objects. This is not surprising in retrospect: if the SSL representation is indeed largely invariant to shifts, i.e. to object position, then the optimal deterministic reconstruction for a given representation is a pixel-space average of the object at all positions: an awful blur. The mapping form input to SSL representation should be thought of as a many-to-one mapping, so inverting it must be one to many.
>
> IC-GAN without neighborhood sampling may be a good alternative worth exploring indeed, as could be other generative models s.a. a conditional VAE or conditional normalizing flow. We primarily chose to use a diffusion approach because they lately seemed the state-of-the-art generative modeling approach for image-quality. Our contribution is not to tell what is the best generative model to use, but what one can see and learn by tweaking a current s.o.t.a. generative model as a tool to analyse the representations learned by SSL approaches.
>
> *"Diffusion models do not usually explicitly express an energy function" (Footnote 4) The usual training mechanism is to train the reweighted epsilon-parameterization from [5] which corresponds to a denoised score matching objective. This score corresponds to the derivative of an energy which can be plugged into MCMC sampling, see [6].”*
>
> Indeed, but what is typically trained is a parameterized score predictor, while the energy function itself (whose derivative this score corresponds to) is usually not expressed explicitly. This is what we meant with this footnote remark, which -granted- is incomplete to do justice to the range of theoretical development and foundations laid out for these methods in the literature. We replaced it by an appropriate list of references.
>
> *“The figure on the first page should show several examples of each model to emphasize the visualization capability of the invariances.”*
>
> The complete figure that shows the invariances is at the end of the appendix (Figure 34). We didn’t put it in the front due to space constraints.
>
> *“The interpolation in Fig. 4c should show several "interpolation-paths", demonstrating the stochasticity of the proposed approach (DDPM)”*
>
> We added Figure 16 in the appendix.
>
> *“Does the proposed conditioning mechanism via conditional batch norm provide significant advantages over other methods (e.g. addition to the timestep-embedding, AdaIN, cross-attention, ...)?”.*
>
> This is a very good question. Our main focus was on analysing representations, so we stuck with the first approach that worked well (and is also well established and of common use in the literature). But refining and evaluating the pros and cons of different conditioning mechanisms is an interesting future direction.

---

> > ### Comment · Reviewer_rQaS · 2021-11-26
> > **Thanks**
> >
> > Thank you for the detailed response and the update on the paper. I think the restructuring has improved the paper overall and I am raising my score to 6.

---

> ### Author Response · Authors · 2021-11-23
> **Answer to reviewer rQaS (2/3)**
>
>
> *“ Further, I do not quite understand the focus on SSL representations, as the proposed techniques can easily be applied to any type of representation, independent of the training paradigm.”*
>
> Certainly, a conditional generative model could be used to “visualize” the information retained  in any kind of representation, and we also compare with supervised representations in the paper.
> SSL representations are of particular interest because they are relatively new, and their properties little explored and poorly understood. Their training criteria is supposed to enforce specific invariances, and it is unclear to what degree they achieve this, and what else they focus on. Currently, researchers in SSL will use toy datasets or artificial datasets based on mnist to evaluate how much their training criterion makes their representation invariant to a given factor of variation. However, the conclusion made on these toy datasets will not necessarily translate on real images dataset like ImageNet (and labelling each factor of variation from ImageNet will be too expensive). Thus, the message behind our work is to show to the SSL community that there is a very simple method which can give us meaningful insight about representations without the need of a labeled dataset. Furthermore SSL methodes use data augmentation to enable their training, this enforces the need of being able to verify the invariances with respect to these augmentation.
> It’s true that our technique can be used for any type of representation but again, our contribution is to showcase that having  such a model enables novel, powerful and telling analysis of self-supervised learning of representations.
>
>
> *“Moreover, a quantitative assessment of the correspondences between representations and synthesized images is missing. For example, can the synthesized representations be fed back into the (SSL) model and does this then yield the same representation.”*
>
> Thank you for suggesting shedding more light on this aspect. Generated inputs fed back into the SSL encoder will be close, but usually do not yield a numerically-exact identical representation. Also the absolute numerical value of distances between them in representation space is not very meaningful as a comparison metric, since it depends e.g. on the arbitrary scale of the representation space. So we tried to define a useful global-scale-insensitive retrieval-based metric. We compute the representation associated with an image, then we feed this representation to RCDM to generate many samples. We compute the representation of each of the generated samples and compute which are the closest neighbors in the representation space within the full validation set of ImageNet of these samples. If the closest neighbor is the representation of the image used as conditioning, it means that the samples have their representations that are very very close to the original image. This allows us to compute the rank of the original conditioning within the generated samples. We see in new Figure 2, table b) that the mean rank is essentially 1 for all of the SSL models while it is much higher for the supervised model.

---

> ### Author Response · Authors · 2021-11-23
> **Answer to reviewer rQaS (1/3)**
>
> Thank you for your time and your careful review. Please first read our “common answer” above. We now answer your specific comments:
>
> *“The main drawback of this paper, however, is that this idea is not novel and the paper fails to discuss prior work which uses different implementations for the conditional generative model, e.g. GANs [1], autoregressive models [2] or normalizing flows [3]. Authors should clearly situate their work within this existing literature and discuss what advantages/differences an approach based on DDPMs offers. For example, is it possible to use a deterministic approximation of the diffusion process (such as DDIM, see [4]) to extract, analogously to [3], an explicit representation of the learned invariances?”*
>
> Thank you for pointing out relevant prior work that we missed. We added a discussion of these in the last paragraph of Section 2, contrasting them with our approach. We restate these contrasts in more details below. Also the point and main contribution of our work is not the specific conditional generative model we built and use (while we think we are the first to use this specific combination of components, it is hardly original), but to showcase how one can use a high quality conditional generative model to gain novel insights into SSL representations, and how they differ from supervised ones.
>
> We understand the connection between the normalizing flow used in [3] and the DDIM process however the focus of this work is not to learn a mapping between representations. In contrast with [3], and to be coherent with the SSL setup, we don’t use any labelled attributes. Also we don’t want to disentangle the representation towards specific factors of variations, but merely to analyse the representation in the form it is given. In contrast with [3], we show that a simple stochastic decoder can enable image manipulation from the representation space without the need of labelled factors of variation. This result highlights that there is already some structure that occurs in SSL representations (as showcased in Section 5.3 and section E.2 of the paper) which doesn’t seem to be “present” or easy to catch in supervised representations.
>
> A key motivation for us to use a diffusion model, beyond the above point, is that conditional diffusion model offer state of the art generation, and thus seems adequate given our goal.

---

### Official Review · Reviewer_MCTj · 2021-11-04

**Correctness:** 3
**Technical Novelty And Significance:** 3
**Empirical Novelty And Significance:** 2
**Recommendation:** 5
**Confidence:** 4

**Main Review:**

Strengths:

1. The question is meaningful, and the proposed technique is interesting.
2. Visualizations generated using the proposed technique look promising in addressing the question laid out in the study.
3. The paper is very well-written, and the analyses are well-motivated.

Critiques:
1. Some conclusions are stated as though they are specific to SSL but it might be good to know whether or not they are applicable for supervised training as well. For instance, the authors claim that `mapping to the same SSL representation as a natural image is not sufficient for producing a similar realistic-looking image’. My intuition is that that should be true for the supervised case as well due to the very nature of gradient-based matching on a deep layer representation when starting from random noise images. This result is not very surprising and has been described before.

2. Imposing the 'naturalness’ constraint always arises out of a need for human understandability but it is not the most faithful account of what information is contained in the representation. I am a little concerned by the use of the phrase 'high fidelity’ in this setting to describe the visualization/model inversion process.  Perhaps it would be useful to briefly discuss the effect of different constraints on the input space in terms of fidelity – one with fewer control knobs (regularizations, naturalness prior etc.) is going to rank better on the fidelity scale..?  And the perceptual quality of image generation is not likely a good reflection of its faithfulness to the representation.


3. Given that the goal was to characterize (through visualization) what is contained in a self-supervised representation, I think the results are not particularly intriguing and seem inadequate in their present form. Additional work and more rigorous analyses, including exploring more models, producing stronger results by analyzing more images etc., seem needed to strengthen the paper.

4. The proposed method for analyzing representations is also highly subjective (not unlike other interpretability approaches I concede) but still there are no recommendations or thorough analysis about how to find the key factors driving a representation. e.g., how many images should be sampled from the RCDM to find the commonalities? Further, the conclusions that are drawn using this conditional image syntheses approach could just as well have been drawn by analyzing the representations directly (for instance, by seeing how well the scale of images can be decoded from the representation).


**Summary Of The Paper:**

This paper proposes a representation analysis technique to particularly understand what information is contained in self-supervised models. The authors propose a visualization technique based on conditional diffusion models that seeks to synthesize realistic-looking inputs whose representation matches that of a target image. Finding commonalities among these synthesized images likely provides clues about the target representation.

**Summary Of The Review:**

I enjoyed reading the paper. However, I'm uncertain about how much value this paper adds in efforts towards understanding the nature of self-supervised representations and the results are not very interesting. As a result, I think the paper falls slightly below the acceptance threshold.

---

> ### Author Response · Authors · 2021-11-23
> **Answer to reviewer MCTj (2/2)**
>
>
> *“Given that the goal was to characterize (through visualization) what is contained in a self-supervised representation, I think the results are not particularly intriguing and seem inadequate in their present form. Additional work and more rigorous analyses, including exploring more models, producing stronger results by analyzing more images etc., seem needed to strengthen the paper”*
>
> We already provided and analyzed numerous diverse images in the initial appendix, and we added additional experiments in the updated version of the paper. We also intend to release the code as opensource and to release checkpoints, to allow anyone to generate any desired image based on their application at hand. Concerning the suggestion of exploring more models, we highlight that we already have Dino, SimCLR, Barlow T, Swav, VicReg and a supervised model. Concerning the SSL models, we also trained RCDM on the projector, for each of these models. And for Dino, we also trained RCDM with different image resolutions.
>
> *”The proposed method for analyzing representations is also highly subjective (not unlike other interpretability approaches I concede) but still there are no recommendations or thorough analysis about how to find the key factors driving a representation. e.g., how many images should be sampled from the RCDM to find the commonalities? Further, the conclusions that are drawn using this conditional image syntheses approach could just as well have been drawn by analyzing the representations directly (for instance, by seeing how well the scale of images can be decoded from the representation).”*
>
> Regarding the point about the key factors driving a representation, we invite you to read the new section 5.3 in the updaded paper as well as the section E.2 in the appendix (Especially Figure 6, 31, 32). We presented a simple heuristic to find the most and least common key factors in representations and visualize through RCDM what those factors correspond to. We were successfully able to identify factors such as the dimensions associated to the background as well as the ones associated to some clothing.
>
> We disagree that the conclusions drawn using our conditional image synthesis could all just as easily be drawn by analyzing the representations directly. While one could, as you rightly suggest, try to decode the scale of the main object from the representation, this requires having a dataset with labeled scales. And similarly for all aspects one would want to probe, which can be complicated or costly to obtain. Having a natural image dataset of the size of ImageNet with all interesting factors of variation labeled in each image would require a tremendous amount of work. It is also conceivable to probe for information pertaining specific transformations we know how to apply (s.a. scaling). But this also will be limited, to a set of simple hand-coded transformations (how to probe for 3d rotation angle?). By contrast visualizing synthesized images has the potential to visually reveal all aspects that are stable (thus encoded) and all that vary (thus not encoded), even if we cannot easily label or code what we are looking to probe for. Or even when we don’t know what to look for beforehand!
>
> Another unique possibility is showcased in fig 6, 31, 32, 33. Even if we don’t know how to manipulate inputs, we instead can manipulate representations, and see the impact of the conditionally generated images. One can perform algebraic manipulation of the representation, or separately manipulate (groups of) dimensions to shed light on the role of each.

---

> ### Author Response · Authors · 2021-11-23
> **Answer to reviewer MCTj (1/2)**
>
> Thank you for your time and your careful review. Please first read our “common answer” above. We now answer your specific comments:
>
> *“Some conclusions are stated as though they are specific to SSL but it might be good to know whether or not they are applicable for supervised training as well. For instance, the authors claim that `mapping to the same SSL representation as a natural image is not sufficient for producing a similar realistic-looking image’. My intuition is that that should be true for the supervised case as well due to the very nature of gradient-based matching on a deep layer representation when starting from random noise images. This result is not very surprising and has been described before.”.*
>
> We decided to shift the gradient-based matching to the appendix as it is incidental to the central message of our work. As you mention, this is a result that was already known and also holds in the supervised case. We hope the updated version clarifies and better highlights the unique contributions of this work (see e.g. the last paragraph of the introduction).
>
> *“Imposing the 'naturalness’ constraint always arises out of a need for human understandability but it is not the most faithful account of what information is contained in the representation. I am a little concerned by the use of the phrase 'high fidelity’ in this setting to describe the visualization/model inversion process. Perhaps it would be useful to briefly discuss the effect of different constraints on the input space in terms of fidelity – one with fewer control knobs (regularizations, naturalness prior etc.) is going to rank better on the fidelity scale..? And the perceptual quality of image generation is not likely a good reflection of its faithfulness to the representation.”*
>
> In our paper when we wrote about “High Fidelity” we meant how realistic the samples generated are, thus how “fidel” are the samples with respect to the manifold of natural images. To clarify, we now define and use the term “faithful” when we refer to how close the representation of generated samples is to the representation used for conditioning. With these definitions, we want to emphasize that we don’t claim that our method is the best one to get the most faithful samples with respect to the representation (we however compare favorably in faithfulness with an IC-GAN model provided by its authors, see comparison we added in Fig. 24). Yet, we present a method that gives us a good trade-off in terms of Fidelity (w.r.t. the manifold of natural images, as per Tab 2a) and Faithfulness to a given representation (as per Tab 2b).

---

### Author Response · Authors · 2021-11-23
**Common answer to reviewers**

We would like to start by thanking all reviewers for their careful reading of our paper, and for their thoughtful comments, which helped us improve the paper and, we hope, clarify its message.
The main and common issue highlighted in the reviews was that the contribution this research makes was unclear. We thus want to clarify for all reviewers that:

a) We do not consider the conditional diffusion model we develop and use as a particularly noteworthy technical contribution. Even if to our knowledge we are the first to use this specific architectural combination (as it suited our needs), it is a, hardly original, combination of preexisting building blocks.

b) This paper is also not about claiming superiority, or comparing amongst multiple alternative possible conditional generative approaches (diffusion, GAN, normalizing flow, VAE, autoregressive, ...). We primarily opted for a denoising-diffusion based approach, as they currently seem state-of-the-art amongst generative models in generated image quality. Why should we use a different approach?

c) Our aim and primary contribution is to showcase how such a high quality conditional generative model can be used to shed new light on what aspects a given representation (SSL or supervised) retains or doesn’t retain about the input. This e.g. allows us to visually compare the representations of self-supervised models and supervised models, showing clear qualitative differences in what they encode and are invariant to. Our focus on self-supervised learning is motivated by the desire to better understand what these approaches actually manage to learn -- which is still poorly-understood -- and by the potential of our method for visually revealing all a representation knows about. This, even in the absence of labels (thus consistent with SSL training) or specific annotated factors of variations to probe for (expensive to collect, especially on natural images).

We updated the paper to clarify and strengthen the aspect of probing SSL representations in multiple ways.

List of changes to the paper:
- To more clearly highlight the contributions, we updated the abstract, the end of the introduction (contributions), and the conclusion.
- We moved the section about gradient optimization over the input space to the appendix, as it is tangential to our primary contribution. We also moved Fig 3 (now fig 11 in appendix) that describes the model, since diffusion models are already well detailed in cited papers.
- We revamped our contributions to clearly identify the conclusions that can be reached from visual inspections of the generated samples (page 2).
- We improved the related work (page 4) to include and discuss works pointed out by reviewers. Thanks!
- We added quantitative measurements of how representation-faithful our model is. That is, how close are the representations of generated images from the images’s representations used for conditioning (Table/Figure 2b and Fig 18 and 24), also compared with IC-GAN.
- We added an experiment to further investigate the difference between supervised model and SSL models under adversarial attacks in Sect 5.2 (Fig. 5 and Fig. 30). This demonstrates yet another crucial difference between both frameworks.
- We added Sect 5.3 as an exploratory analysis on the information contained in each dimension of SSL models (Fig. 6, 31, 32).

List of the new figures: Fig 5,6,16,18,19,20,24,28,30,31,32.

Previous figure 1 2 3 are moved to appendix fig 7 8 11.

Fig 5b is completely removed.

Current Fig 1 is updated (previously known as Fig 4).

New table in Figure 2.

---

### Decision · Program_Chairs · 2022-01-20

**Decision:**

Reject

**Comment:**

This paper proposes a method for visualizing representations of neural networks trained with self-supervised learning with conditional denoising diffusion probabilistic models. By generating multiple images conditioned on a representation, one can identify what aspects the representation is and is not sensitive to. The proposed method allows for high fidelity generated images that can be used to compare different self-supervised methods and layers.

Reviewers agreed that the paper proposed reasonable methodology, targeted an interesting problem of understanding what is learned by self-supervised methods, and presented interesting qualitative evaluations. However, there remained concerns on the novelty of results in comparison to other methods for probing representations (e.g. classification based), subjectiveness of interpretation of the qualitative results, and limited quantifications of the intuition gained from the visualizations. While the authors have argued that the point of the paper is to showcase the merits of qualitative visual analysis method, reviewers found that the presented results were insufficient to demonstrate the value of the proposed approach. A number of ideas were discussed with reviewers on how to highlight the value of visualization which could strengthen the paper in the future. Given the lack of novelty on the conditional generation side, and limited insight gained from the qualitative results, I cannot recommend this paper for acceptance in its current form.